# Neuromodulation of striatal D1 cells shapes BOLD fluctuations in anatomically connected thalamic and cortical regions

**Marija Markicevic[1,2,3]\*, Oliver Sturman[2,4], Johannes Bohacek[2,4], Markus Rudin[5,6], Valerio Zerbi[7,8†], Ben D Fulcher[9†], Nicole Wenderoth[1,2,10]\*†**

[1]Neural Control of Movement Lab, HEST, ETH Zürich, Zurich, Switzerland; [2]Neuroscience Center Zurich, University and ETH Zurich, Zurich, Switzerland; [3]Department of Radiology and Biomedical Imaging, School of Medicine, Yale University, New Haven, United States; [4]Laboratory of Molecular and Behavioral Neuroscience, Institute for Neuroscience, HEST, ETH Zurich, Zurich, Switzerland; [5]Institute of Pharmacology and Toxicology, University of Zurich, Zurich, Switzerland; [6]Institute for Biomedical Engineering, University and ETH Zurich, Zurich, Switzerland; [7]Neuro-X Institute, School of Engineering (STI), EPFL, Lausanne, Switzerland; [8]CIBM Centre for Biomedical Imaging, Lausanne, Switzerland; [9]School of Physics, The University of Sydney, Camperdown, Australia; [10]Future Health Technologies, Singapore-ETH Centre, Campus for Research Excellence and Technological Enterprise (CREATE), Singapore, Singapore

**\*For correspondence:**
marija2m4@gmail.com (MM);
nicole.wenderoth@hest.ethz.ch
(NW)

†Shared last authorship

**Competing interest:** The authors declare that no competing interests exist.

**Abstract** Understanding how the brain's macroscale dynamics are shaped by underlying microscale mechanisms is a key problem in neuroscience. In animal models, we can now investigate this relationship in unprecedented detail by directly manipulating cellular-level properties while measuring the whole-brain response using resting-state fMRI. Here, we focused on understanding how blood-oxygen-level-dependent (BOLD) dynamics, measured within a structurally well-defined striato-thalamo-cortical circuit in mice, are shaped by chemogenetically exciting or inhibiting D1 medium spiny neurons (MSNs) of the right dorsomedial caudate putamen (CPdm). We characterize changes in both the BOLD dynamics of individual cortical and subcortical brain areas, and patterns of inter-regional coupling (functional connectivity) between pairs of areas. Using a classification approach based on a large and diverse set of time-series properties, we found that CPdm neuromodulation alters BOLD dynamics within thalamic subregions that project back to dorsomedial striatum. In the cortex, changes in local dynamics were strongest in unimodal regions (which process information from a single sensory modality) and weakened along a hierarchical gradient towards transmodal regions. In contrast, a decrease in functional connectivity was observed only for cortico-striatal connections after D1 excitation. Our results show that targeted cellular-level manipulations affect local BOLD dynamics at the macroscale, such as by making BOLD dynamics more predictable over time by increasing its self-correlation structure. This contributes to ongoing attempts to understand the influence of structure–function relationships in shaping inter-regional communication at subcortical and cortical levels.

## Editor's evaluation

This manuscript provides a valuable set of findings that provide clarity on how striatal direct pathways regulate macroscopic information flow in the brain via the thalamus. The manuscript represents a powerful piece of evidence that will be relevant to researchers across many fields in neuroscience.

## Introduction

The brain is a complex network of anatomically connected and perpetually interacting neural components. Perturbing an individual component at the cellular level can alter both the local neural dynamics and patterns of inter-regional communication. A large body of work has used functional magnetic resonance imaging at rest (rsfMRI) to investigate interregional functional connectivity (FC) at the macroscale with cellular manipulations (*Rocchi et al., 2022*; *Markicevic et al., 2020*; *Zerbi et al., 2019*) and without (*Chuang and Nasrallah, 2017*; *Gozzi and Schwarz, 2016*; *Grandjean et al., 2020*). But relatively few studies have focused on directly capturing the local dynamical properties of blood-oxygen-level-dependent (BOLD) signal fluctuations *within* brain areas (*Wang et al., 2023*). How microscale alterations shape macroscale dynamics is key to understanding the complex brain dynamics.

One approach to tackle complex interactions that underlie BOLD fluctuations is to understand the influence of structural connections and structure–function relationships. Structural connections have been shown to constrain FC, with the strength of this effect varying along a unimodal-to-transmodal cortical gradient in humans: structure–function coupling is higher for unimodal regions that primarily process information from a single sensory modality than for transmodal regions that integrate information across modalities of large-scale, polysynaptic circuits (*Bazinet et al., 2021*; *Margulies et al., 2016*; *Vázquez-Rodríguez et al., 2019*). This hierarchical unimodal–transmodal gradient is reflected in cortical microstructural properties, such as cytoarchitecture, size, density and distribution of dendritic cell types, synaptic structure, gene expression, and macroscale FC in humans and non-human primates (*Burt et al., 2018*; *Huntenburg et al., 2018*). Using an MRI-derived biomarker, that is, T1-weighted/T2-weighted ratio (T1w/T2w), which acts as a highly correlated proxy for the structural cortical hierarchy in *Burt et al., 2018*, *Fulcher et al., 2019* have shown that cortical gradients also exist in mouse and correspond to approximate hierarchical gradients in cytoarchitecture, interneuron cell density, long-range axonal connectivity, and gene expression. However, it remains unknown whether regional BOLD dynamics also vary along this gradient. Recent work has indicated that local BOLD dynamics are shaped by structural connectivity, with more strongly connected regions exhibiting slower timescales of BOLD dynamics in mouse (*Sethi et al., 2017*) and human (*Fallon et al., 2020*). Time-series properties of the BOLD signal also vary with molecular, cellular, and circuit properties of the brain (*Gao et al., 2020*; *Shafiei et al., 2020*). While these correlational studies provide an understanding of the relationships between the anatomical properties of brain areas and their BOLD dynamics, direct manipulations are required to provide causal evidence for how cellular-level properties shape BOLD dynamics.

In our previous work, we used rsfMRI with chemogenetic neuromodulation of the somatosensory cortex in mice and measured systematic alterations in BOLD dynamics of the manipulated brain area (*Markicevic et al., 2020*). Resulting changes to the statistical properties of BOLD time series in individual brain areas were quantified using a machine-learning approach that leverages thousands of time-series features (*Fulcher and Jones, 2017*; *Markicevic et al., 2020*). This approach of combining thousands of time-series features (including linear autocorrelation, entropy and predictability, and outlier properties) is termed 'highly comparative time-series analysis' (*Fulcher et al., 2013*). This approach is available as a software tool, *hctsa*, which provides a data-driven way to assess and interpret changes in a wide variety of possible time-series properties (*Fulcher and Jones, 2017*). Here we apply this tool to regions forming a well-defined striato-thalamo-cortical circuit to understand (i) how a cell-specific perturbation of a striatal region affects the local BOLD dynamics within the manipulated area and within its projection areas (i.e., structurally connected regions in thalamus and cortex) and (ii) which structural or functional properties of the projection areas constrain the influence exerted by modulating striatal activity.

To explore this, we chemogenetically excited, or inhibited, the activity of D1 medium spiny neurons (MSN) in the right dorsomedial caudate putamen (CPdm)—the 'input area' of the striatum (*Hintiryan et al., 2016*; *Lee et al., 2016*; *Runegaard et al., 2019*; *Wall et al., 2013*)—while acquiring rsfMRI data across the whole brain (*Figure 1A*). We investigated how this chemogenetic manipulation shaped regional BOLD signal fluctuations (*Figure 1B*) in the right CPdm-thalamo-cortical loop as assessed via a large set of time-series properties (*Fulcher and Jones, 2017*; *Fulcher et al., 2013*) and a multivariate classification algorithm to evaluate changes (*Figure 1A*). We found that alterations of D1 MSN activity in the CPdm shape the BOLD dynamics of multiple right thalamic and cortical regions. These dynamical changes were strongest for (i) thalamic subregions which project back to neuromodulated

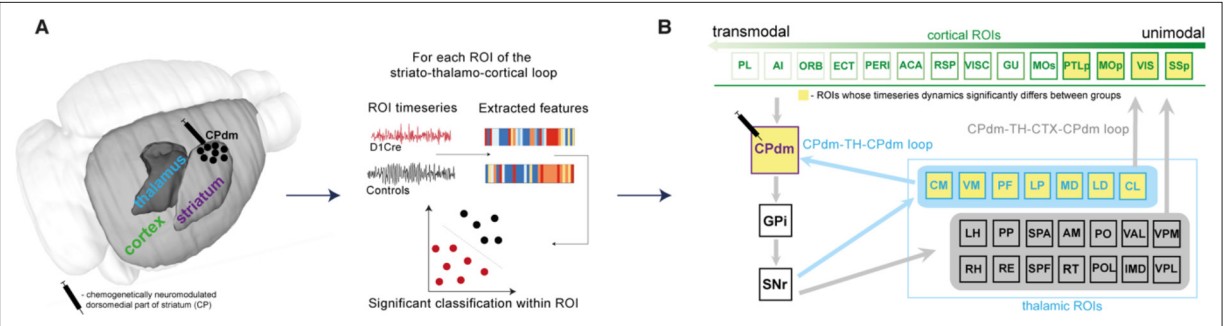

**Figure 1.** The effect of D1 medium spiny neuron (MSN) neuromodulation within dorsomedial striatum was studied within an anatomically defined striato-thalamo-cortical circuit using blood-oxygen-level-dependent (BOLD) time-series properties. (**A**) MSNs of the right dorsomedial striatum (CPdm) were neuromodulated via DREADDs. Time series were extracted from regions of interest (ROIs) in striatum (CP), thalamus (TH), and cortex (CTX) after the DREADDs were activated by clozapine. BOLD time-series properties were computed using the hctsa feature set (*Fulcher and Jones, 2017*). For each ROI, a linear support vector machine was used to determine whether neuromodulated animals could be distinguished from controls using time-series features extracted from the BOLD signal. (**B**) ROIs within the striato-thalamo-cortical circuit connected to CPdm were identified using mesoscale anatomical atlases. Thalamic areas were subdivided into ROIs that project back to CPdm (light blue box and light blue arrows) versus thalamic ROIs that do not (light gray box). Cortical ROIs (green) were ranked along a previously described multimodal hierarchy (*Fulcher et al., 2019*). ROIs marked in yellow indicate that neuromodulated versus control animals could be significantly distinguished based on BOLD time-series properties. In cortex, such ROIs are primarily unimodal, while in thalamus such ROIs project back to the neuromodulated CPdm. Full names of abbreviated ROIs: AI, agranular insular area; PL, prelimbic area; ECT|PERI, ectorhinal and perirhinal area; VISC|GU, visceral and gustatory areas; ORB, orbital area; RSP, retrosplenial area; MOp/s, primary and secondary motor cortex; SSp, primary somatosensory cortex; ACA, anterior cingulate area; PTLp, posterior parietal association area; VIS, visual area; CPdm, caudate putamen dorsomedial; SNr, substantia nigra reticular part; GPi, globus pallidus internal; VPL, ventral posterolateral nucleus of thalamus; RE|LH|RH, nucleus of reuniens|lateral habenula|rhomboid nucleus; PO|POL, posterior complex and posterior limiting nucleus of thalamus; SPF|SPA|PP, subparafascicular nucleus subparafascicular area with peripeduncular nucleus of thalamus; IMD, intermediodorsal nucleus of thalamus; RT, reticular nucleus of thalamus; AM, anteromedial nucleus; CL, central lateral nucleus of thalamus; PF, parafascicular nucleus; VAL|VPM|VPMpc, ventral anterior-lateral complex of the thalamus with ventral posteromedial nucleus of the thalamus and its parvicellular part; MD, mediodorsal nucleus of thalamus; LD, lateral dorsal nucleus of thalamus; LP, lateral posterior nucleus of thalamus; VM|CM, ventral and central medial nuclei of thalamus.

right CPdm and (ii) unimodal rather than transmodal cortical areas as summarized in *Figure 1B* (significantly affected regions highlighted in yellow). Our results provide a comprehensive understanding of how targeted cellular-level manipulations affect both local dynamics and interactions at the macroscale showing the influence of structural characteristics of a circuit and putative cortical hierarchy in shaping dynamics of cortical and subcortical regions.

## Results

### Chemogenetic manipulation of D1 MSNs of CPdm alters the motor behavior of animals

Four weeks after surgery, mice underwent a behavioral open-field test after activating the DREADDs with a low dose of clozapine (control animals were also injected with clozapine prior to the behavioral test) (*Figure 2B*). The behavioral open-field test was performed as a manipulation check to test whether activating the DREADDs in CPdm caused behavioral changes as predicted by previous work (*Bay König et al., 2019*; *Lee et al., 2016*). Exciting D1 MSNs with clozapine increased the number of contraversive rotations and decreased the number of ipsiversive rotations compared to controls who also received clozapine (MANOVA, $F_{contra}(1,13) = 19.7$, $p_{contra}=0.001$; $F_{ipsi}(1,13) = 8.8$, $p_{ipsi}=0.01$; *Figure 2E and F*). Exciting D1 MSNs also increased the total distance moved (MANOVA, $F(1,13) = 18.2$, $p=0.001$; *Figure 2D*). By contrast, inhibiting D1 MSNs of CPdm decreased the number of contraversive rotations and increased the number of ipsiversive rotations when compared to control mice or to excitatory D1 MSN mice (MANOVA, $F_{contra}(1,19) = 14.7$, $p_{contra}=0.001$; $F_{ipsi}(1,19) = 66.1$, $p_{ipsi}=1 \times 10^{-5}$; *Figure 2E and F*). These results replicate the behavior which is typically observed for unilateral excitation versus inhibition of D1 MSNs (*Bay König et al., 2019*; *Lee et al., 2016*; *Runegaard et al., 2019*; *Tecuapetla et al., 2014*).

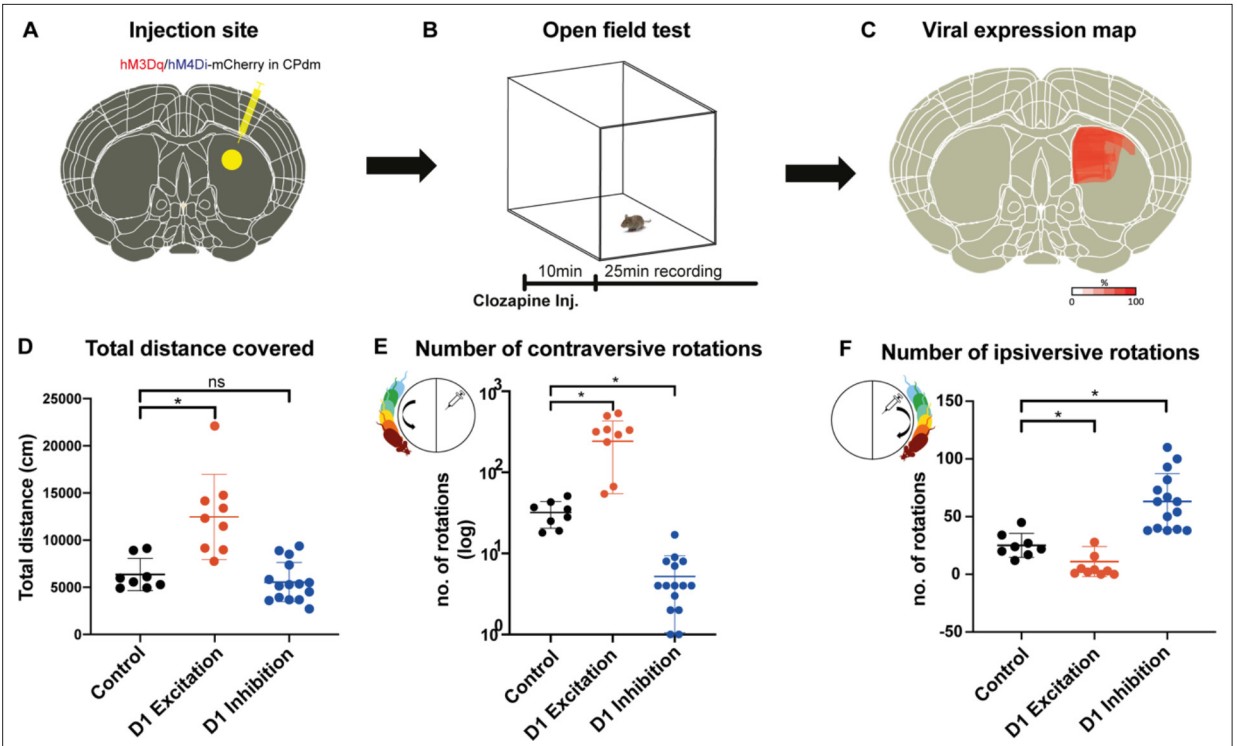

**Figure 2.** Chemogenetic neuromodulation of D1 medium spiny neurons (MSNs) altered the animals' motor behavior. (**A**) Either DIO-hM3Di-mCherry, DIO-hM4Dq-mCherry, or DIO-mCherry (control) virus was injected in the right dorsomedial striatum (CPdm) of D1Cre mice (marked in yellow). (**B**) Four weeks after viral injection, an open-field test was performed. A 25 min recording of mouse behavior was commenced 10 min after intraperitoneal clozapine injection. (**C**) Qualitative viral expression maps were overlaid for all mice included in the experiment (100% on the scale bar indicates the presence of viral expression in all the mice). (**D**) Mice whose D1 MSNs of CPdm were excited and not inhibited covered significantly more distance when compared to controls (MANOVA, p=0.001). (**E**) Exciting D1 MSNs of the right CPdm increased the frequency of contraversive rotations (turning in the direction opposite the injection site as illustrated within the small circle, p=0.001) when compared to controls. On the contrary, inhibiting D1 MSNs of the right CPdm decreased contraversive rotations relative to controls (p=0.001). (**F**) Compared to controls, the number of ipsiversive rotations (turning in the same direction as the injection site, as illustrated within the small circle) significantly decreased when D1 MSNs of the CPdm were excited (p=0.011), while rotations significantly increased when D1 MSNs were inhibited (p=$1.3 * 10^{-5}$).

The online version of this article includes the following figure supplement(s) for figure 2:

**Figure supplement 1.** No difference in number of transfected neurons between excitatory and inhibitory DREADD transfected animals.

We also assessed the viral transfection in the CPdm by immunostaining. *Figure 2C* shows the superimposed viral expression maps of all mice which clearly cover CPdm. These results indicate that our approach successfully modulated D1 MSNs in CPdm. *Figure 2—figure supplement 1* gives information about the viral expression for individual animals. Additionally, using confocal images from mice that underwent D1 excitation and D1 inhibition, we further showed that the number of transfected neurons between DREADD groups was not significantly different (*Figure 2—figure supplement 1B*, t(8) = 0.6; p=0.5).

## Altered dynamics of virally transfected CP region

Seven days after behavioral testing, mice were lightly anesthetized and spontaneous brain activity was measured via rsfMRI before and after activating the DREADDs with clozapine (control animals also received a similar dose of clozapine as DREADD transfected animals). Since we neuromodulated D1 MSNs in the intermediate portion of the right dorsomedial part of CP (CPdm), we were specifically interested in whether this target area would be affected.

To test this, we parcellated the CP based on the viral expression maps obtained from all animals. Specifically, we used the CP parcellated atlas containing 29 sub-regions from *Hintiryan et al., 2016* and overlaid each with viral DREADD expression maps obtained from each animal. We identified which striatal subregions were transfected with the virus using the criterion that viral expression was

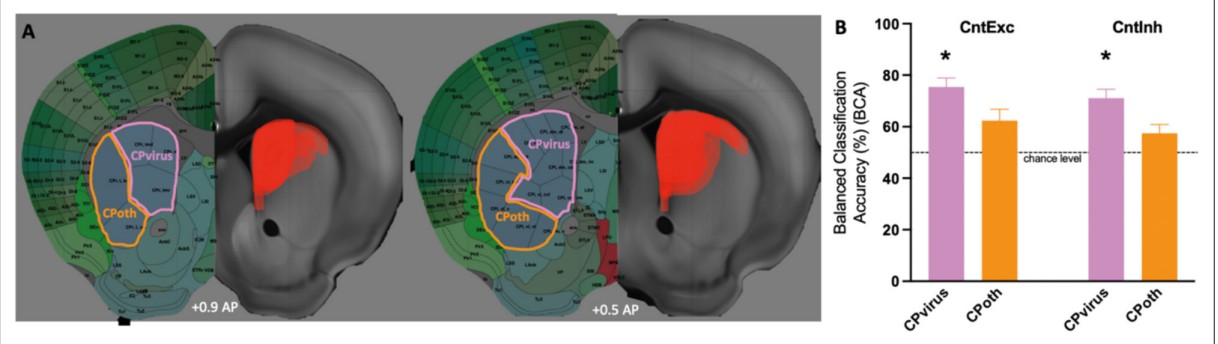

**Figure 3.** Altered dynamics of virally transfected CP subregions. (**A**) Viral spread on the right side of the brain slice. Left side illustrates which CP subregions obtained from Hintiryan atlas were virally transfected by the DREADD virus and form a part of CPvirus region. (**B**) Classification results for CPvirus region and CPoth (region formed from CP subregions that were not transfected by the virus) for the two group comparisons (Controls vs. D1 excitation and controls vs. D1 inhibition). * refers to significance of CPvirus region from chance obtained using permutation testing.

detected in at least 1/4 of the CP sub-area in at least two animals. This analysis revealed that Cpi,d-m,dl; Cpi,dm,dt; Cpi,dm,im; Cpi,dm,cd; Cpi,vm,vm; Cpi,vl,cvl; Cpr,m; Cpr,imd; and Cpi,r,imv have been transfected and these subregions were combined into one region of interest (CPvirus). Next, we extracted BOLD time series from CPvirus and classified controls versus either D1 excitation or D1 inhibition. The balanced classification accuracy for D1 control vs D1 excitation was 75.4% with *p*-value of 0.02, while for D1 control vs D1 inhibition was 71.1% with *p*-value of 0.03 (***Figure 3A-B***). By contrast, when we repeated the same approach for the non-transfected subareas of the CP (CPoth), the changes in classification accuracy were not significant. This finding confirms that the dorsal striatum (our general target area) was transfected by the virus and affected by excitation and inhibition.

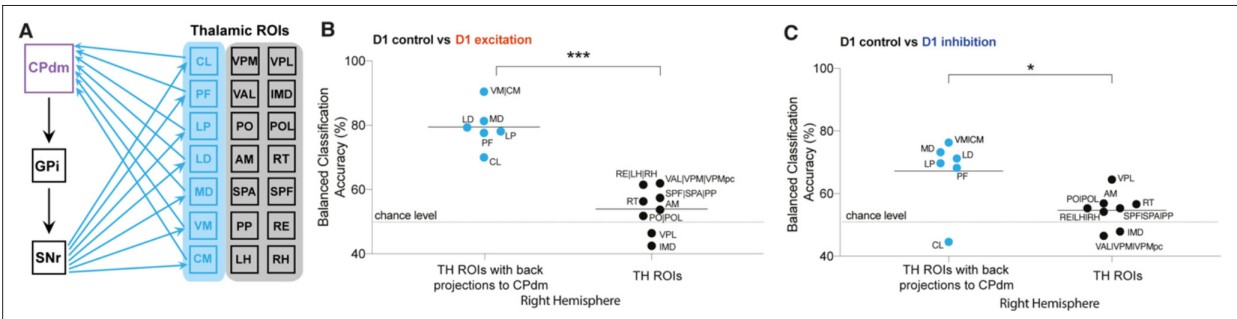

**Figure 4.** Altered blood-oxygen-level-dependent (BOLD) dynamics in thalamic regions forming anatomical loops with dorsomedial striatum. (**A**) Figure illustrating thalamic (TH) subregions (light blue) that project back to dorsomedial striatum. (**B**) Comparison of balanced classification accuracies for D1 control versus D1 excitation group between thalamic regions projecting back to the modulated site CPdm (blue dots) and the rest of the thalamic regions of interest (ROIs) which constitute striato-thalamo-cortical circuit (two-tailed Mann–Whitney *U*-test; p=7 × 10⁻⁴). Thalamic regions which project back to CPdm have significantly higher balanced classification accuracies as compared to the other thalamic ROIs. (**C**) Similar to (**B**) but for D1 control versus D1 inhibitory group comparison (two-tailed Mann–Whitney *U*-test; p=0.04). Full names of abbreviated ROIs: CPdm, caudate putamen dorsomedial; SNr, substantia nigra reticular part; GPi, globus pallidus internal; VPL, ventral posterolateral nucleus of thalamus; RE|LH|RH, nucleus of reuniens|lateral habenula|rhomboid nucleus; PO|POL, posterior complex and posterior limiting nucleus of thalamus; SPF|SPA|PP, subparafascicular nucleus subparafascicular area with peripeduncular nucleus of thalamus; IMD, intermediodorsal nucleus of thalamus; RT, reticular nucleus of thalamus; AM, anteromedial nucleus; CL, central lateral nucleus of thalamus; PF, parafascicular nucleus; VAL|VPM|VPMpc, ventral anterior-lateral complex of the thalamus with ventral posteromedial nucleus of the thalamus and its parvicellular part; MD, mediodorsal nucleus of thalamus; LD, lateral dorsal nucleus of thalamus; LP, lateral posterior nucleus of thalamus; VM|CM, ventral and central medial nuclei of thalamus.

The online version of this article includes the following figure supplement(s) for figure 4:

**Figure supplement 1.** Characteristic changes observed in blood-oxygen-level-dependent (BOLD) dynamics of regions part of the striato-thalamo-cortical circuit.

**Figure supplement 2.** Figure illustrates anatomical projections between dorsolateral (CPdl) and dorsomedial striatum (CPdm) and thalamic and cortical areas that exhibited significantly altered time-series dynamics in response to our experimental manipulation.

## Changes in BOLD dynamics in thalamic subregions that form anatomically closed loops with CPdm

We next sought to understand whether activating the DREADDs in CPdm affects the local BOLD dynamics of anatomically connected thalamic nuclei. In each thalamic region of interest (ROI), we applied a classification analysis to the MSN D1 excitation versus control group and the MSN D1 inhibition versus control group. We found statistically significant classification accuracies ($p_{corr}$<0.05) for multiple thalamic ROIs, that is, parafascicular nuclei (PF), lateral dorsal (LD), lateral posterior (LP), mediodorsal (MD), and ventral and central medial (VM|CM) nuclei of thalamus. Interestingly, the largest effects were observed for those thalamic ROIs that have a specific reciprocal anatomical connection with CPdm (*Figure 4A*, *Figure 4—figure supplement 1A–C*, *Figure 4—figure supplement 2*), that is, they receive projections from CPdm via GP/SN and project back to CPdm. Indeed, ROIs with reciprocal anatomical connections to CPdm displayed higher balanced accuracies than other thalamic ROIs for both D1 excitation versus D1 control (two-tailed Mann–Whitney *U*-test, $U(8,6) = 0$, $z = -3.10$, p=7 × 10$^{-4}$, *Figure 4B*), and D1 inhibition versus D1 control (two-tailed Mann–Whitney *U*-test, $U(8,6) = 8$, $z = -2.07$, p=0.04, *Figure 4C*). In summary, we found that modulating D1 MSN activity alters BOLD signal dynamics in downstream thalamic regions that form strong closed-loop anatomical connections with the neuromodulated CPdm.

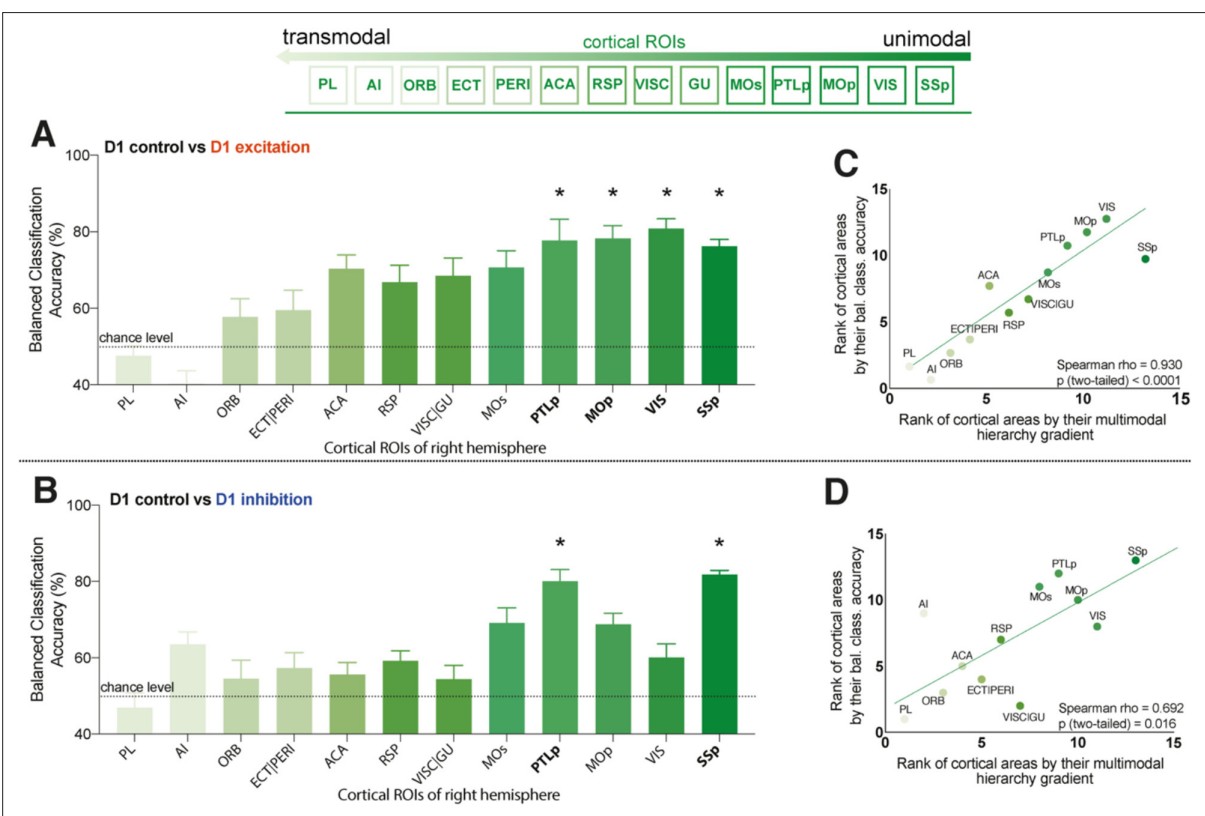

**Figure 5.** Cortical regions of interest (ROIs) with altered blood-oxygen-level-dependent (BOLD) dynamics after D1 medium spiny neuron (MSN) modulation are primarily unimodal. (**A, B**) Balanced classification accuracy (%) results for each cortical brain region shown for: (**A**) D1 control versus D1 excitation; (**B**) D1 control versus D1 inhibition. Regions are listed following a previously described hierarchical ordering of cortical areas (*Fulcher et al., 2019*) that putatively orders regions along a unimodal-to-transmodal axis (illustrated above the two figures). A false discovery rate (FDR)-corrected significant difference in BOLD dynamics between the two groups is depicted with bold ROI abbreviations and an asterisk (*) ($p_{corr}$<0.05). (**C**) Spearman correlation ($\rho$ = 0.93, p=1 × 10$^{-4}$) between cortical areas ranked by their balanced classification accuracy and their multimodal hierarchy gradient for D1 control versus D1 excitation group comparison. (**D**) Same as (**C**) but for D1 control versus D1 inhibition group comparison (rho = 0.69, p=0.02). Full names of abbreviated ROIs: AI, agranular insular area; PL, prelimbic area; ECT|PERI, ectorhinal and perirhinal area; VISC|GU, visceral and gustatory areas; ORB, orbital area; RSP, retrosplenial area; MOp/s, primary and secondary motor cortex; SSp, primary somatosensory cortex; ACA, anterior cingulate area; PTLp, posterior parietal association area; VIS, visual area.

## Cortical regions with altered BOLD dynamics are primarily unimodal

Next, we evaluated the balanced classification accuracy among cortical ROIs during D1 MSN CPdm neuromodulation. Using our classification approach, we assessed changes in BOLD time-series properties of individual cortical areas for D1 excitation versus no modulation in control mice. Significant changes in BOLD dynamics ($p_{corr}<0.05$, permutation test) were observed for four cortical ROIs: SSp, primary somatosensory area; VIS, visual area; MOp, primary motor area; and PTLp, posterior parietal association area (*Figure 5A*). Repeating the analysis for D1 inhibition versus control resulted in two significant regions: PTLp and SSp (*Figure 5B*).

We aimed to understand whether the observed differences in response to D1 MSN manipulation across cortical areas were related to other sources of heterogeneity across cortical areas. In particular, recent work has shown a key spatial dimension of heterogeneity across mouse cortical areas, in which gene-expression patterns, cell-type densities, and other structural properties follow a common, putatively hierarchical variation from primary unimodal areas (like somatomotor areas, which process information from a single sensory modality) to transmodal association areas (which process information from multiple modalities) (*Fulcher et al., 2019*). Combining information from normalized measures of cell densities, cytoarchitecture, T1w:T2w, and brain-relevant gene-expression markers, *Fulcher et al., 2019* used principal components analysis to extract an ordering of cortical areas from transmodal (lowest rank) to unimodal (highest rank). We assessed whether this hierarchical ordering of cortical areas was related to the variation in their dynamical response to D1 MSN neuromodulation in CPdm. The relationship between cortical ROIs ranked according to the Fulcher et al. multimodal hierarchy, and by balanced accuracy computed here, is shown in *Figure 5A and B*. We find a positive correlation between the two measurements for both types of modulation: (i) D1 excitation versus D1 control (Spearman's $\rho$ = 0.93, p=0.0001, *Figure 5C*) and (ii) D1 inhibition versus D1 control (Spearman's $\rho$ = 0.69, p=0.02, *Figure 5D*). Our results indicate that D1 MSN modulation alters a cortical area's BOLD dynamics with a strength that closely follows its hierarchical position, with the strongest response in unimodal areas and the weakest response in transmodal areas.

Similar analyses were performed to explore the changes in the dynamics of cortical and thalamic regions between D1 excitation and D1 inhibition (*Figure 4—figure supplement 1C*). None of the balanced classification accuracies were statistically significant after correction for multiple comparisons, indicating that activation versus deactivation does not result in significantly distinguishable changes in BOLD dynamics.

## Common time-series properties drive successful classification among thalamic regions

Our results demonstrate that changes in macroscopic BOLD dynamics in response to D1 MSN neuromodulation of CPdm can be assessed using a feature-based time-series classification approach. But which properties of the BOLD time series are specific to each manipulation? To answer this, we analyzed the cortical (SSp, PTLp, MOp, and VIS), thalamic (PF, LP, LD, MD, and VM|CM), and striatal (sub)regions where the classification accuracy (using the full *hctsa* feature set) was statistically significant after. For each of these regions, we determined which *individual* time-series features differ significantly between the D1 excitation versus D1 control group by calculating the signed Mann–Whitney rank-sum statistic and associated p-value (for each individual feature) and correcting for multiple comparisons across all features (n = 6588) by controlling the false discovery rate (FDR). Repeating the analysis described above for the significant regions (PTLp and SSp) after D1 inhibition was compared to D1 control did not find any individually significant features. Therefore, we focus on the results after D1 excitation was compared to D1 controls and refer to this Mann–Whitney rank-sum statistic as a 'feature score' here.

Discriminative individual features were found in two regions: (i) VM|CM (344 significant features) and (ii) LD (284 significant features). These two regions have the highest balanced classification accuracies using the full *hctsa* feature set (*Figure 4—figure supplement 1A*): (i) 90.4% and (ii) 80.3%. First, we focused on VM|CM to assess which types of BOLD time-series properties features best distinguish D1 excitation from D1 control by manually inspecting the list of 344 significant features (*Supplementary file 2*). Overall, we found that these high-performing features were directly measuring or sensitive to changes in various types of autocorrelation properties of the BOLD signal. To investigate this observation further, we focused on the 100 most discriminative features, which have $p_{FDR}<0.01$

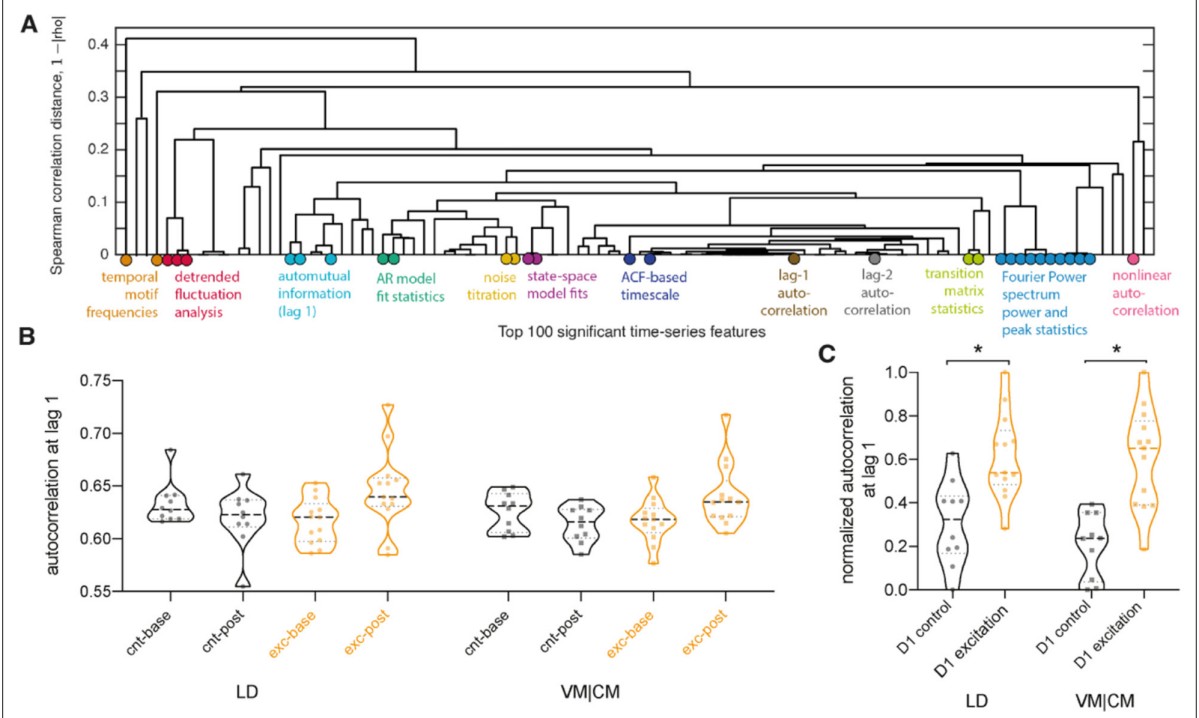

**Figure 6.** D1 medium spiny neuron (MSN) excitation led to slower and more autocorrelated fluctuations of the time-series signal in thalamic regions. (**A**) The dendrogram captures the similarity between the top 100 features at distinguishing the stimulation condition from blood-oxygen-level-dependent (BOLD) dynamics in VM|CM. These top 100 features were selected by those with the strongest difference between stimulation and control conditions according to a Mann–Whitney rank-sum statistic. Feature dissimilarity is measured as Spearman correlation distance (1-|rho|) between all pairs of features which were calculated across all animals and subjected to clustering. The dendrogram organizes features using hierarchical average linkage clustering. Selected features are annotated; refer to *Figure 6—figure supplement 2A* for a full annotation of all features onto the dendrogram. (**B**) Autocorrelation at lag 1 shown for baseline and post clozapine controls (cnt-base, cnt-post, respectively) and excitatory groups (exc-base, exc-post) for lateral-dorsal (LD) and ventral-, central-medial (VM|CM) thalamic regions. Increase in autocorrelation is observed during a post clozapine period in the excitatory group, for both brain regions (LD and VM|CM). (**C**) Increase in autocorrelation is observed in normalized within-group baseline-corrected data. *indicates a between groups false discovery rate (FDR)-corrected significant difference, that is, $p_{LD}$=0.03; $p_{VM|CM}$=0.009. Full names of abbreviated regions of interest (ROIs): LP, lateral posterior nucleus of thalamus; VM|CM, ventral and central medial nuclei of thalamus.

The online version of this article includes the following figure supplement(s) for figure 6:

**Figure supplement 1.** Verified correct implementation of cross-validation and no overfitting using high-dimensional feature space.

**Figure supplement 2.** Broadly similar changes in time-series properties after D1 excitation in significant thalamic, cortical, and striatal regions.

(note that the patterns are similar when analyzing all 344 significant features). *Figure 6A* visualizes each of these 100 time-series features as a leaf in a dendrogram, where features with similar behavior on the dataset (measured as the absolute Spearman correlation, $|\rho|$) are placed close to each other. Selected features are annotated onto the dendrogram using colored circles (for a detailed annotation of all features, see *Figure 6—figure supplement 2A*). The dendrogram reveals a range of time-series features, derived from different types of methods, that exhibit high overall similarity to each other as measured by the Spearman correlation distance ($1-|\rho| \leq 0.43$) on this dataset: from low time-lag linear autocorrelation coefficients, correlation timescale estimates, state-space and AR model fits, properties of the Fourier power spectrum, automutual information, and fluctuation-analysis methods. Through different specific algorithmic formulations, these features all capture the increase in self-correlation of the signal—consistent with slower and more autocorrelated fMRI signals—in the D1 stimulation condition relative to control. This is most clearly exhibited by the example of lag-1 linear autocorrelation, AC(1), displayed in *Figure 6B and C*. Overall, our analysis demonstrates that many individual methods from across time-series analysis—which are sensitive to differences in autocorrelation structure (including low-lag autocorrelations and various autocorrelation timescales)—are useful for distinguishing D1 stimulation from VM|CM fMRI time series.

Next, we aimed to explore whether the same types of BOLD time-series properties vary in broadly similar ways for the LD (for which individual features differed significantly between the D1 excitation and D1 control group); and the remaining regions that exhibited high-classification accuracies using the full *hctsa* feature set. In short, we assessed whether changes in BOLD dynamics due to the different stimulation conditions are similar to changes observed in VM|CM by computing correlations between the feature scores (signed rank-sum statistics) for each pair of regions (*Figure 6—figure supplement 2B and C* and shown below). To assess the significance of computed correlation coefficients, we developed a permutation-based method to compute a p-value (using 5000 correlated group-label shuffles; i.e., permuting 'stimulation-on' and 'stimulation-off' labels, while maintaining consistent labeling across the two regions). That is, for each pair of regions, we estimated a null distribution using 5000 group-label shuffles. We found high ($\rho > 0.4$) and significant correlations between feature scores between VM|CM and other cortical, striatal, and thalamic (sub)regions, indicating that D1 excitation causes broadly similar changes to BOLD time-series properties in those regions.

As above, we computed a Mann–Whitney rank-sum test statistic score of each feature for each region and correlated these feature scores with those obtained for VM|CM. The statistical significance of these correlation coefficients was estimated using the permutation test developed above. These p-values (11 for the correlation to each other region) were corrected to control the FDR at 0.05. Results indicate that positive correlation coefficients between VM|CM and other thalamic, cortical, and striatal regions, which range from $\rho$ = 0.38–0.58, are significant (*Figure 6—figure supplement 2C*). A specific illustration of this finding is shown in *Figure 6—figure supplement 2D* explicitly, showing the increase in autocorrelation at lag 1 after stimulation for the other thalamic and cortical regions. Overall, these results point towards broadly similar changes in time-series properties after D1 excitation in significant thalamic, cortical, and striatal regions.

## Multiple cortical regions display changes in functional connectivity after CPdm neuromodulation

Since excitation and inhibition of D1 MSNs of CPdm significantly alter the local dynamics of remote brain regions in the striato-thalamo-cortical circuit, we next examined whether their pairwise coupling, measured as FC using linear Pearson correlations, was also affected (*Figure 7*). We assessed how injecting clozapine changes FC between CPdm and each of the anatomically connected ROIs.

First, animals from all three groups were pooled and baseline FC (i.e., before the DREADDs were activated with clozapine) was calculated. One-sample Wilcoxon tests revealed that ACA, PL, MOp, RSP, GPe, RT, LP, and LD were positively correlated with CPdm, while GPi, VAL|VPM, and VM|CM were negatively correlated ($p_{corr}<0.01$, FDR-corrected, *Figure 7A*). Next, we investigated how FC changed from baseline to the post-clozapine period for each of the three groups and each ROI. For example, FC between CPdm and ACA (*Figure 7B*) was reduced when D1 MSNs were excited (*Figure 7B*, red trace), while it was increased when D1 MSNs were inhibited (*Figure 7B*, blue trace), and it remained largely unchanged in the control group (*Figure 7B*, black trace).

Direct comparison between D1 excitation and controls indicated a significant decrease in FC after D1 excitation between CPdm–ACA and CPdm–MOp connections ($p_{corr}<0.05$, FDR-corrected, *Figure 7C*). Additionally, when we compared whether exciting versus inhibiting D1 MSNs differentially affects CPdm connectivity, we saw that excitation typically reduced FC, while there was a trend towards increased FC during inhibition (*Figure 7C*). This was statistically significant for FC changes between CPdm and ACA, and RSP and MOp ($p_{corr}<0.05$, FDR-corrected). We also performed an analysis of DREADD-induced FC changes among all ROIs of striato-thalamo-cortical circuit, including thalamostriatal and thalamo-cortical connections, but found no significant connections ($p_{corr}<0.05$, FDR-corrected; data not shown). Overall, our FC results indicate a significant decrease in FC between CPdm and three cortical nodes when D1 MSNs were excited and compared to either controls or D1 inhibition. To ensure that this significant result is not influenced by FC changes in control animals due to clozapine injection, we re-ran the FC analysis between CPdm and ACA and MOp without normalization. *Figure 7—figure supplement 1* shows that most of the control animals retain nearly identical raw FCs from baseline to post clozapine, while there was a clear change in FC for the D1 excitation and D1 inhibition groups.

We further investigated whether there is any relationship between individual behavioral result (post clozapine) and pairwise coupling (FC) between CPdm and ACA, MOp, SSp, and PTLp. These regions

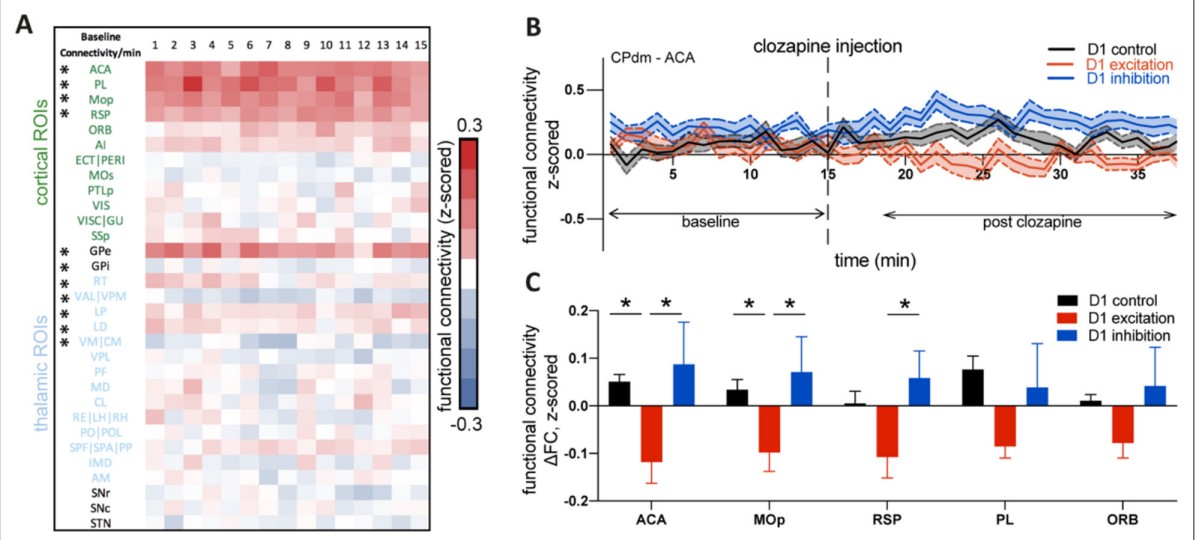

**Figure 7.** Cortico-striatal changes in functional connectivity (FC) upon D1 medium spiny neuron (MSN) neuromodulation. (**A**) Baseline FC calculated per minute between CPdm and the rest of regions of interest (ROIs) that form the striato-thalamo-cortical loop, combining animals from all three groups involved in the experiment. One-sample Wilcoxon test was performed, and * indicates that statistical significance was reached after false discovery rate (FDR) correction for multiple comparisons ($p<0.01$). (**B**) FC calculated by minute for D1 controls, D1 excitation, and D1 inhibition groups between CPdm and ACA (an example ROI). Dashed line at 15 min indicates clozapine injection, that is, activation of the DREADD, which is followed by the changes in FC for D1 excitation and D1 inhibition groups. The baseline (first 15 min before clozapine injection) and post clozapine periods (from 19 to 34 min) used for the delta analysis (**C**) are indicated. (**C**) Functional connectivity at ΔFC (post clozapine – baseline) for cortical regions that displayed differences during pairwise comparisons (permutation test, $p_{uncorr}<0.05$). * indicates that statistical significance was reached after FDR correction on all pairwise comparisons of all three groups. Full names of abbreviated ROIs: AI, agranular insular area; PL, prelimbic area; ECT|PERI, ectorhinal and perirhinal area; VISC|GU, visceral and gustatory areas; ORB, orbital area; RSP, retrosplenial area; MOp/s, primary and secondary motor cortex; SSp, primary somatosensory cortex; ACA, anterior cingulate area; PTLp, posterior parietal association area; VIS, visual area; CPdm, caudate putamen dorsomedial; SNr, substantia nigra reticular part; GPi, globus pallidus internal; VPL, ventral posterolateral nucleus of thalamus; RE|LH|RH, nucleus of reuniens|lateral habenula|rhomboid nucleus; PO|POL, posterior complex and posterior limiting nucleus of thalamus; SPF|SPA|PP, subparafascicular nucleus subparafascicular area with peripeduncular nucleus of thalamus; IMD, intermediodorsal nucleus of thalamus; RT, reticular nucleus of thalamus; AM, anteromedial nucleus; CL, central lateral nucleus of thalamus; PF, parafascicular nucleus; VAL|VPM|VPMpc, ventral anterior-lateral complex of the thalamus with ventral posteromedial nucleus of the thalamus and its parvicellular part; MD, mediodorsal nucleus of thalamus; LD, lateral dorsal nucleus of thalamus; LP, lateral posterior nucleus of thalamus; VM|CM, ventral and central medial nuclei of thalamus.

The online version of this article includes the following figure supplement(s) for figure 7:

**Figure supplement 1.** Functional connectivity (FC, corresponding to the z-scored Pearson's correlation coefficient) between Cpdm-ACA (upper row) and CPdm-MOp (lower row) separately displayed for D1 excitation, control, and inhibition at the baseline level and after clozapine injection for each animal individually (in the same graph the same-colored dots represent the same animal).

**Figure supplement 2.** Significant correlation between rotational behavior and edges with significantly altered functional connectivity after neuromodulation.

were chosen based on (i) the strength of the FC results (**Figure 7C**); (ii) strong structural connectivity with CPdm as described by **Hintiryan et al., 2016**; and (iii) high balanced classification accuracy using *hctsa*. First, we subtracted the number of ipsiversive rotations from the number of contraversive rotations for each animal and used this as a metric of the behavioral effect measured 10 min after subcutaneous clozapine injection. The behavioral metric was then correlated with the animal's individual FC metric. We found that animals which exhibit strong alterations of FC between CPdm and ACA or MOp (**Figure 7—figure supplement 2A and B**) also exhibit strong behavioral changes ($r \geq 0.681$, $p<0.0001$). This was only true for cortical regions whose FC is strongly altered after D1 neuromodulation but not for regions with limited FC changes, even if they reached high classification accuracy (**Figure 7—figure supplement 2C and D**, $r \leq 0.054$, $p \geq 0.768$). This establishes an association between the post-clozapine behavior in an individual mouse and changes in FC between CPdm and the primary motor cortex or anterior cingulate area.

## Discussion

Understanding how the brain's macroscale dynamics are shaped by underlying microscale mechanisms is a key problem in neuroscience. Here, we used an anatomically well-defined striato-thalamo-cortical circuit to investigate how cell-specific chemogenetic excitation or inhibition of D1 MSNs of right dorsomedial striatum affects regional BOLD dynamics and pairwise coupling. Using a classification approach based on a large and diverse set of time-series properties from the *hctsa* toolbox (*Fulcher and Jones, 2017*), we found that the CPdm neuromodulation alters BOLD signal dynamics in anatomically connected sub-areas of thalamus and cortex. Importantly, the regional changes in BOLD time-series properties were stronger for (i) thalamic regions with reciprocal, structural connectivity with CPdm; and (ii) cortical areas situated lower in a recently proposed multimodal hierarchy. We found that D1 excitation increases the autocorrelation (and low-frequency power) of the BOLD signal, showing how neuromodulation shapes the temporal autocorrelation of distributed but connected regions. We also assessed FC between the neuromodulated dorsomedial striatum and areas of the connected striato-thalamo-cortical circuit, but only observed decreased striato-cortical FC after D1 MSN excitation. Taken together, we show that fMRI time-series properties can be used to study how the dynamics within macroscopic brain regions are affected by targeted cellular-level neural manipulations. Our results suggest that regional BOLD dynamics vary across the hierarchy, with unimodal regions being more sensitive to perturbations from dorsomedial CP. Overall, these findings show that targeted cellular-level manipulations affect local BOLD dynamics at the macroscale, such as by making BOLD dynamics more predictable over time by increasing its self-correlation structure. This contributes to ongoing attempts to understand the influence of structure–function relationships in shaping inter-regional communication at subcortical and cortical levels.

### Altered BOLD dynamics in thalamic regions mediated by direct anatomical projections with CPdm

The thalamus is a highly heterogeneous structure, forming bidirectional connections with visual, limbic, associative, sensory, and motor regions of the cortex, as well as striatum. Extensive research has focused on anatomically mapping the striato-thalamic and thalamostriatal projections (*Alloway et al., 2017*; *Antal et al., 2014*; *Bubb et al., 2017*; *Díaz-Hernández et al., 2018*; *El-Boustani et al., 2020*; *Elena Erro et al., 2002*; *Guo et al., 2015*; *Hunnicutt et al., 2016*; *Kamishina et al., 2008*; *Lee et al., 2020*; *Linke et al., 2000*; *Mandelbaum et al., 2019*; *Namboodiri et al., 2016*; *Parker et al., 2016*; *Perry and Mitchell, 2019*; *Smith et al., 2004*; *Van der Werf et al., 2002*; *Vertes et al., 2015*; *Wall et al., 2013*; *Wang et al., 2006*), which enabled us to further distinguish thalamic ROIs based on their projections to/from the neuromodulated CPdm. We show that after D1 MSN excitation and inhibition, only those thalamic ROIs that form anatomically closed loops with CPdm display altered BOLD dynamics. Correlational analyses from *Sethi et al., 2017* have shown that regional BOLD dynamics are related to the strength of the incoming projections, where stronger incoming projections displayed more autocorrelated dynamics. Our CPdm manipulations resulted in BOLD signal perturbations (assessed by significant balanced classification accuracies) only for thalamic regions with anatomical projections originating in the CPdm. These thalamic regions also displayed more autocorrelated dynamics after neuromodulation, providing causal evidence in support of the strength of incoming projections shaping regional BOLD dynamics. Our findings contribute to ongoing attempts to understand the influence of structural connections, as a function of their strength, in shaping inter-regional communication among cortical and subcortical regions. The results presented here provide evidence for the causal mechanisms that are needed to accurately develop, refine, and validate quantitative models of the brain's spatially distributed dynamics (*Breakspear, 2017*).

### Cortical regions with altered BOLD dynamics are primarily unimodal

Out of 12 cortical regions included in the striato-thalamo-cortical circuit, 4 exhibited significantly altered BOLD dynamics after D1 MSN excitation and 2 after D1 MSN inhibition. Several studies have provided evidence that stimulation in thalamus, cortex and striatum can also spread to generate alterations in BOLD signals of remote regions through multiple synapses (*Lee et al., 2016*; *Matsui et al., 2012*). Our results provide further evidence that modulating D1 MSN activity in the CPdm seems to cause changes in BOLD dynamics which propagate through poly-synaptically connected circuits.

There was substantial variation in how strongly the BOLD dynamics of cortical regions changed in response to neuromodulating striatal D1 MSN cells. Our results indicate that this variation follows a recently described multimodal hierarchical gradient (*Fulcher et al., 2019*). This gradient orders cortical areas along a putative functional hierarchy from unimodal primary somatosensory areas to transmodal integrative prefrontal areas using diverse measurements of cortical microstructural properties, such as cytoarchitecture, cell-type densities, and brain-relevant gene expression levels, as well as noninvasive MRI-derived T1w/T2w measures, axonal input strength. The results indicate that macroscale dynamics of individual regions exhibit a tightly coupled spatial variation with diverse measurements of cortical microstructure, which follow a common gradient of variation. This common gradient may provide insights into why the dynamics of nominally unimodal cortical areas are most sensitive to D1 neuro-modulation of CPdm. Compared to transmodal regions, unimodal regions have weaker axonal input strength (*Fulcher et al., 2019*) and fewer integrative functions (*Wang, 2020*; *Wang, 2022*), which are responsible for integrating inputs from systems implicated in perceiving and acting. This may explain their greater sensitivity to the CPdm perturbation. Furthermore, *Sethi et al., 2017* have shown that local BOLD dynamics are correlated to the strength of anatomical projections.

Diverse measurements of cortical microstructural properties form macroscopic gradients, argued to shape brain dynamics and function (*Parkes et al., 2022*; *Siegle et al., 2021*; *Wang, 2020*; *Wang, 2022*). For example, *Burt et al., 2018* show that parvalbumin (PV+) neurons are twice as abundant in unimodal areas than association/transmodal areas, while the opposite is true for somatostatin (SST) neurons (*Burt et al., 2018*; *Wang, 2020*). *Kim et al., 2017* have demonstrated that PV/SST cell densities strongly vary across a proposed hierarchy of mouse cortical areas, with sensory areas containing a greater proportion of putatively output-modulating PV neurons and, conversely, transmodal areas containing a greater proportion of putatively input-modulating SST neurons. This is consistent with the hypothesis that unimodal regions are more involved in feed-forward information streams that require more output-modulating SST neurons than input-modulating PV neurons. Some supporting evidence for this hypothesis comes from *Fulcher et al., 2019*, who found a weak but significant relationship between T1w:T2w (a candidate proxy for hierarchical level; *Burt et al., 2018*) and the total incoming structural projection strength to a cortical area, with a lower aggregated input weight to unimodal compared to transmodal areas. In this emerging picture of unimodal areas as being involved in fast processing of sensory inputs and receiving relatively fewer inputs than transmodal areas, a constant perturbation would be predicted to have a greater effect on the dynamics of a unimodal area (for which it constitutes a larger perturbation relative to the low baseline level of input) than a transmodal area (for which it constitutes a relatively small perturbation to the large baseline level of input). This general prediction is supported by our results, in particular a very strong correlation with the response to striatal stimulation across the cortical hierarchy ($\rho > 0.7$). That is, we found that the change in BOLD dynamics is strongest in unimodal areas and becomes progressively more subtle along the hierarchy. However, given the range of characteristics that vary across cortical areas and may play a role in shaping regional dynamics (*Shafiei et al., 2023*; *Shafiei et al., 2020*), future investigations will be crucial in uncovering the relationship between diverse properties of cortical structure and their influence in shaping macroscale neural dynamics of individual cortical area.

The change in time-series properties observed in cortical areas low in the hierarchy (putatively 'unimodal') is an intriguing result, particularly because primary motor (MOp) and somatosensory cortex (SSp) have been shown to project to the dorsolateral (CPdl) rather than the dorsomedial striatum (CPdm). We argue that these changes are likely to be driven via a striato-thalamo-cortical route. As discussed above, dorsomedial but not the dorsolateral striatum has dense, reciprocal connections to the identified thalamic nuclei. Importantly, nearly all of these nuclei project to the unimodal regions, including MOp and SSp, as summarized in *Figure 4—figure supplement 2*. Thus, striato-thalamo-cortical projections do not seem to exhibit the same level of segregation as cortico-striatal projections, and we argue that the effect of our experimental manipulation on time-series dynamics has propagated via this pathway. It is therefore likely that intrinsic properties of unimodal cortical regions result in an increased sensitivity of their BOLD signal to these striato-thalamic inputs.

## D1 MSN excitation leads to slower and more autocorrelated BOLD signal fluctuations

An increase in autocorrelation of the BOLD time series was observed for two significant thalamic regions following D1 excitation. The time-series features responsible for successful classification indicate an increase in self-correlation structure in the fMRI signal. Thorough analyses of individual features that distinguish the stimulation condition ranged from linear to nonlinear autocorrelation coefficients, automutual information, statistics of Fourier power spectrum, and others. Our analyses show that these features were correlated to different types of autocorrelation properties, pointing towards a common change in fMRI signal: an increase in its self-correlation structure. These results are broadly consistent across regions constituting striato-thalamo-cortical circuit. They are also in line with the widespread fMRI literature investigating fMRI time series using different statistics derived from the linear autocorrelation function or Fourier power spectrum (*Fallon et al., 2020*; *Nakamura et al., 2020*; *Sethi et al., 2017*; *Shinn et al., 2021*; *Wang et al., 2023*; *Zou et al., 2008*). Specifically, *Nakamura et al., 2020* detected an increase in the fractional amplitude of low-frequency fluctuations (fALFF) after excitation of D1 MSNs in the dorsal CP, which is in line with our own results that revealed an increase in low-frequency power, albeit in thalamic regions. Moreover, *Shafiei et al., 2020* report that regional differences in temporal autocorrelation of the human cortex are shaped by molecular and cellular properties of cortical regions. Following this notion, our results provide causal evidence that cellular excitation of D1 MSNs shapes the temporal autocorrelation of anatomically connected regions at the macroscale.

The mechanisms behind increased autocorrelation within regions of striato-thalamo-cortical circuitry after D1 excitation remain unexplored. Assessment of neuronal spike-count autocorrelation has shown that different regions exhibit varied timescales of fluctuations in neuronal population activity, that is, magnitude of autocorrelation (*Churchland et al., 2011*; *Murray et al., 2014*), which varied in the presence of a stimulus or a task (*Nougaret et al., 2021*; *Runyan et al., 2017*; *Wasmuht et al., 2018*). Using calcium imaging, increased autocorrelation has been observed in parietal cortex neurons following an auditory decision-making task in mice (*Runyan et al., 2017*), indicating that autocorrelation may be a statistical property that enables information coding by neuronal populations over different timescales. The variability of the magnitude of the autocorrelation of intrinsic neural signal was also demonstrated using other neuroimaging modalities such as resting-state fMRI data and human intracranial recordings (*Gao et al., 2020*; *Stephens et al., 2013*; *Watanabe et al., 2019*). Based on this, we speculate that D1 CPdm neuromodulation alters neuronal autocorrelation at the population level, shaping the dynamics of the fMRI signal, which is also characterized by an increase in autocorrelated fluctuations. This could mean autocorrelation is a statistical property that is informative of changes observed across different spatial scales of a dynamical system, that is, from micro- to macro-scales. For a detailed mechanistic understanding of changes in autocorrelation across different spatial scales, future multimodal experiments combining fMRI with invasive measurements, such as calcium imaging in animal models, as well as computational modeling are necessary.

## Multiple cortical regions display changes in functional connectivity after CPdm neuromodulation

We demonstrate differential modulation of striato-cortical FC between CPdm and the rest of the striato-thalamo-cortical regions. Nakamura and colleagues have performed electrophysiological recordings in both dorsal striatum and motor cortex before and after D1 MSN excitation. Results indicate that neuromodulation increased delta power in dorsal striatum but not in motor cortex (*Nakamura et al., 2020*). Since slow oscillations of delta-band fluctuations contribute to FC (*Wang et al., 2012*) and are characterized by brain-wide synchrony (*Pan et al., 2013*; *Uhlhaas et al., 2010*), we speculate that differences in low-frequency power changes between the neuromodulated CPdm and cortex contribute to our observations of decreased striato-cortical FC after exciting D1 MSNs. Moreover, recent findings illustrate that low power coherence between chemogenetically inhibited areas and remote non-neuromodulated areas also contributes to shaping FC between those regions (*Rocchi et al., 2022*). This indicates that more comprehensive analysis combining electrophysiology with fMRI could provide clearer insights into the mechanisms behind observed changes in FC.

Moreover, we assessed the relationship between rotational behavior and FC changes. The significant correlations establish a relationship at the individual mouse level between observed (post

clozapine) behavior and observed changes in FC in the primary motor cortex, as well as anterior cingulate area. This suggests that these striato-cortical connections are associated with executing coordinated movements (*Klaus et al., 2017*), a finding in line with a number of reports showing that the activity of D1 MSNs in dorsal striatum is necessary for successful execution of locomotion (*Badreddine et al., 2022*; *Barbera et al., 2016*).

## Disconnect between balanced classification accuracy and functional connectivity results

FC captures the correlation of the BOLD time series *between two regions*, while time-series analysis captures systematic changes in how the BOLD signal fluctuates *within a brain region*. Thus, results obtained with the SVM approach versus the FC analyses allow researchers to study neural dynamics through two different lenses which provide rich and complementary information on neural dynamics. In this study, the regions with the strongest changes in FC to other regions were not the same as the regions whose local dynamics were most affected by stimulation. Our FC analysis, using Pearson's correlations between pairs of regions captures how the strength of their contemporaneous linear statistical association changes after D1 neuromodulation. Studies from our prior work have illustrated rather weak connectivity of thalamic regions with cortical and striatal regions obtained from anesthetized mice (*Grandjean et al., 2017*), even after cortical neuromodulation (*Markicevic et al., 2020*). Following this, it is unsurprising that this neuromodulation study does not reveal any FC changes between thalamo-cortical or cortico-thalamic areas. This is one of the reasons we were interested in directly quantifying changes to BOLD signal dynamics in individual regions after D1 neuromodulation. One example of the added value of feature-based classification analyses are the results we have obtained for the thalamus. The properties of the local BOLD signal are conceptually distinct from how they couple to other regions (the alignment of their fluctuations through time) (although they are not strictly independent, e.g., see *Afyouni et al., 2019*; *Cliff et al., 2021* for an account of how the autocorrelation properties of signals affect the resulting null distribution of Pearson correlation coefficients). This suggests the importance of assessing brain function and dynamics using multiple approaches.

It is currently unclear which biological mechanism(s) might underpin changes in FC versus changes in time-series dynamics as detected by our classification analyses, but one potential speculative explanation is that FC changes align with information transfer from one area to another, while changes in time-series dynamics might reflect how one area influences neural processing of another which might be sensitive to the effect of neuromodulators.

## Limitations and interpretational issues

Here we used a comprehensive data-driven approach of classifying BOLD dynamics using a wide range of candidate time-series features. While this approach of using thousands of features was able to capture statistical changes in BOLD dynamics resulting from our experimental manipulation, it limited our ability to identify which individual features are driving the observed changes due to the substantial multiple-hypothesis correction. Future studies might build on our findings and a priori limit the number of features tested (e.g., based on an unsupervised or supervised feature reduction procedure) (*Lubba et al., 2019*). Moreover, our findings could potentially be of benefit for the modeling community to guide the development and validation of useful dynamical models of the brain's distributed dynamics.

Dynamics of the basal ganglia structures downstream from CP, namely GPi/e, SNr/c, and STN, are not affected by exciting or inhibiting D1 MSNs. While this is at odds with conventional models of the basal ganglia, some studies indicate that after D1 excitation or D1 inhibition within the dorsal striatum, the firing rates of these downstream structures change in a non-uniform manner, that is, with partial firing rate increases and partial decreases within the single structure (*Freeze et al., 2013*; *Kravitz et al., 2010*; *Lee et al., 2016*; *Tecuapetla et al., 2014*). In addition, a recent tracer study showed that projections from CP to the output nuclei GPi and SNr are topographically segregated so that CPdm projects only to a specific anatomically defined section of GPi/SNr (*Lee et al., 2020*). If only a small fraction of an already small nucleus is affected by CPdm neuromodulation, the spatial resolution of the BOLD signal is insufficient. This would explain why the BOLD dynamics of regions downstream of striatum are only moderately affected by D1 MSN excitation or inhibition.

We used Hintiryan's cortico-striatal atlas (*Hintiryan et al., 2016*) to parcellate CP into its subregions. These subregions were further inspected for the extent of the viral spread and merged into one large CP region. The fine-grained investigation of individual CP subregions was not performed due to statistical power limitations. Tracer studies indicate anatomically segregated cortico-striatal projections (*Foster et al., 2021*; *Harris et al., 2019*). While it is generally true that dorsal striatum is anatomically segregated into a dorsomedial and dorsolateral portion which receive projections from distinct cortical targets, recent studies using optogenetics have shown that this segregation may not be as strict as initially thought when experimentally stimulating dorsomedial versus dorsolateral portions of the striatum (*Grimm et al., 2021*; *Lee et al., 2016*). For example, *Lee et al., 2016* optogenetically activated D1 MSNs in CPdm (i.e., the same target area as in the present study) and observed strong functional activity in cortical sensorimotor areas. This discrepancy might be explained by striato-thalamo-cortical projection where recent studies have shown that CPdm projects to specific thalamic nuclei which, in turn, project to unimodal cortical areas including primary motor and somatosensory cortex (see *Figure 4—figure supplement 2*). *Bay König et al., 2019* have recently shown that D1 inhibition at the dorsomedial striatum using DREADDs induces rotational behavior, which is directly proportional to the spatial extent of the viral expression. We used rotational behavior to confirm that viral expression of CPdm was successful. Future studies might additionally use individual expression variations to further assess structure–function relationships in a more detailed manner.

## Conclusions

In summary, we characterized the influence of local cellular perturbations on macroscale BOLD dynamics for each region within an anatomically connected system. Our results indicate that alterations of BOLD dynamics are shaped by (i) anatomical connectivity of thalamic subregions with CPdm and (ii) spatial location of cortical regions along a putative hierarchical dimension. Our results show that targeted cellular-level manipulations affect local BOLD dynamics at the macroscale, such as by making BOLD dynamics more predictable over time by increasing its self-correlation structure. This contributes to ongoing attempts to understand the influence of structure–function relationships in shaping inter-regional communication at subcortical and cortical levels. Furthermore, we provide causal evidence into how regional dynamics change after neuromodulation shaping different aspects of the BOLD autocorrelation properties of distributed but connected regions at the macroscale.

## Materials and methods

All experiments and procedures were conducted following the Swiss federal ordinance for animal experimentation and approved by Zurich cantonal Veterinary Office (ZH238/15 and ZH062/18). House inbred BAC-mediated transgenic mouse line from GENSAT (BAC-Cre Drd1a-262–D1Cre) (*Gong et al., 2003*) was used in this study. All D1Cre mice were kept in standard housing under 12 hr light/dark cycle with food and water provided ad libitum throughout the whole experiment. A total of 38 mice were used in the experiment, aged 16.2 ± 2.8 wk and weighing 24.9 ± 3.3 g at the day of the surgery. Prior to surgery, the mice were randomly assigned to any of the three groups, that is, D1 controls, D1 inhibition, or D1 excitation. Following the 3R measures (https://www.swiss3rcc.org/en/3rs-resources/what-are-the-3rs) and based on our previous experience with DREADD experiments (*Markicevic et al., 2020*; *Zerbi et al., 2019*), our sample size goal per group was set to an approximate n = 15 for D1 inhibition D1 excitation and n = 10 for D1 controls.

### Stereotactic transfection procedure

Each mouse was initially anesthetized using a mixture of midazolam (5 mg/mL; Sintetica, Switzerland), fentanyl (50 mg/mL; Actavis AG, Switzerland), and medetomidine (1 mg/mL; Orion Pharma, Finland). After anesthesia induction, mice were placed on a heating pad and the temperature was kept at 35°C (Harvard Apparatus, USA). Following shaving and cleaning, an incision along the midline of scalp was made. The intermediate portion of the right dorsomedial caudate-putamen (CPdm) was targeted at the coordinates of +0.5 mm AP (anterior-posterior), –1.5 mm ML (medio-lateral), and –3.0 mm DV (dorso-ventral) relative to the Bregma using a drill and microinjection robot (Neurostar, Germany) with a 10 µL NanoFil syringe and 34Ga bevelled needle (World Precision Instruments, Germany). Then, 950 nL of double-floxed inverted (DIO) recombinant AAV8 virus was used to express either

hM3Dq-mCherry (excitatory DREADD, n = 13, eight females) or hM4Di-mCherry (inhibitory DREADD, n = 15, seven females) or mCherry (control, n = 10, six females). The virus was injected at the rate of 0.06 uL/min and provided by Viral Vector Core Facility of the Neuroscience Centre Zurich (http://www.vvf.uzh.ch/en.html). After injection, the needle was left in place for 10 min and then slowly withdrawn. Subsequently, mice were given an anesthesia antidote consisting of temgesic (0.3 mg/mL; Reckitt Benckiser AG, Switzerland), annexate (0.1 mg/mL; Swissmedic, Switzerland), and antisedan (0.1 mg/mL; Orion Pharma) and left to fully recover. Following the surgery, ketoprofen (10 mg/kg; Richter Pharma AG, Austria) was subcutaneously injected daily for at least 3 d to reduce any postoperative pain. Animals were given 3–4 wk to fully recover from the surgery and allow for expression of the transgene prior to the scanning session. The viral expression map showing the distribution of the viral expression of all mice included in this study is shown in *Figure 2C* and *Figure 2—figure supplement 1*.

## Behavioral open-field test

A custom-made box (50 × 50 × 50 cm) consisting of light gray walls and a floor was designed and placed in a room with a homogenously spread light source. Each mouse spent 5 min exploring the box before the start of the experiment. To activate the DREADD, clozapine was intraperitoneally injected at a dose of 30 µg/kg, 10 min before the start of the recording. A total of 32 mice underwent an open-field behavioral test and each recording lasted 25 min. Data was analyzed using EthoVision XT14 (Noldus, the Netherlands) software for total distance traveled, and clockwise and anticlockwise rotations for each mouse. Statistical analysis was performed using one-way multivariate ANOVA implemented in SPSS24 (IBM, USA). To account for possible sex differences, age and weight were used as covariates.

## MRI setup and animal preparation

Resting-state fMRI (rsfMRI) measurements were obtained with a 7T Bruker BioSpec scanner equipped with a Pharmascan magnet and a high signal-to-noise ratio (SNR) receive-only cryogenic coil (Bruker BioSpin AG, Fällanden, Switzerland) in combination with a linearly polarized room temperature volume resonator for rf transmission.

Standardized anesthesia protocols and animal monitoring procedures were used when performing rsfMRI scans (*Markicevic et al., 2020*). Briefly, mice were initially anesthetized with 3% isoflurane in 1:4 $O_2$ to air mixture for 3 min to allow for endotracheal intubation and tail vein cannulation. Mice were positioned on an MRI-compatible support, equipped with hot water-flowing bed to keep the temperature of the animal constant throughout the entire measurement (36.6 ± 0.5°C). The animals were fixed with ear bars and mechanically ventilated via a small animal ventilator (CWE, Ardmore, USA) at a rate of 80 breaths per minute, with 1.8 mL/min flow of isoflurane at 2%. Subsequently, a bolus containing a mixture of medetomidine (0.05 mg/kg) and pancuronium (0.25 mg/kg) was injected via the cannulated vein and isoflurane lowered to 1%. Five minutes following the bolus injection, a continuous infusion of medetomidine (0.1 mg/kg/hr) and pancuronium (0.25 mg/kg/hr) was started while isoflurane was further reduced to 0.5%. Animal preparation took on average 16.1 ± 2.7 min, and all animals fully recovered within 10 min after the measurement.

## Resting-state fMRI acquisition and data preprocessing

For fMRI recordings, an echo planar imaging (EPI) sequence was used with the following acquisition parameters: repetition time TR = 1 s, echo time TE = 15 ms, flip angle = 60°, matrix size = 90 × 50, in-plane resolution = 0.2 × 0.2 mm², number of slices = 20, slice thickness = 0.4 mm, 2280 volumes for a total scan of 38 min. Clozapine was intravenously injected 15 min after the scan start at a dose of 30 µg/kg and a total of 38 D1Cre animals (10 controls, 13 D1 excitatory, and 15 D1 inhibitory mice) were scanned.

Data was preprocessed using a previously established pipeline for removal of artifacts from the multivariate BOLD time-series data (*Markicevic et al., 2020*; *Zerbi et al., 2015*). Briefly, each 4D dataset (three spatial plus one temporal dimension) was normalized to a study-specific EPI template (Advanced Normalization Tools, ANTs v2.1, https://picsl.upenn.edu/software/ants/) and transferred to MELODIC (Multivariate Exploratory Linear Optimized Decomposition of Independent Components) to perform a within-subject spatial-ICA with a fixed dimensionality estimation (number of components

set to 60) (*Avants et al., 2011*; *Beckmann and Smith, 2005*). The procedure included motion correction and in-plane smoothing with a 0.3 mm kernel. An FSL-FIX study-specific classifier, obtained from an independent dataset of 15 mice, was used to perform 'conservative' removal of the variance of the artifactual components (*Zerbi et al., 2015*). Subsequently, the dataset was despiked, band-pass filtered (0.01–0.25 Hz) based on the frequency distribution of the fMRI signal under isoflurane-medetomidine anesthesia (*Grandjean et al., 2014*; *Pan et al., 2013*), and finally normalized into the Allen Mouse Brain Common Coordinate Framework (CCFv3) (*Wang et al., 2020*) using ANTs. From each dataset, the first 900 datapoints (equivalent to 15 min of scanning) were used as a baseline. After the clozapine injection, we discarded the following 4 min (or 240 data points) to allow time for the DREADD to become fully activated (*Markicevic et al., 2020*). The subsequent 900 data points (from 19 to 34 min) were analyzed to estimate post-clozapine effects. The difference between baseline and post-clozapine measurements is further referred to as ΔFC (for functional connectivity) or BCA (for balanced classification accuracy – see below for details).

## Defining ROI based on structural connectivity within the striato-thalamo-cortical circuit

The Allen Mouse Brain Connectivity Atlas (AMBCA) (*Oh et al., 2014*) was used to map out the structural connectome of the striato-thalamo-cortical circuit containing our striatal target area CPdm. The mesoscale structural connectome of the mouse brain was derived from 469 viral microinjection experiments in the right hemisphere of C57BL/6J mice. These data were further annotated using the Allen Reference Atlas (*Dong, 2008*) and summarized in the form of weighted, directed connectivity matrix with 213 brain regions obtained using a regression model (*Oh et al., 2014*). The model directly outputs a 'normalized connection strength,' estimated by scaling the injection volume in a source region to explain the segmented projection volume in a target region, and a p-value for each edge in the connectome. This was used to construct a 213 × 213 ipsilateral connectivity matrix. For the purposes of this study, we used the edges (p-value<0.05) of this ipsilateral connectivity matrix to map out regions that constitute the striato-thalamo-cortical circuit.

Based on these structural connectivity data, we reconstructed which ROIs form the striato-thalamo-cortical loop, including CPdm, using a stepwise approach. First, starting from the caudate putamen (CP) as our 'seed area,' we identified globus pallidus external (GPe)/internal (GPi) and substantia nigra pars compacta (SNc)/pars reticulata (SNr) as regions that CP directly projects to. Second, GPe/GPi and SNc/SNr were used as seed areas to identify subthalamic nuclei (STN) and several thalamic ROIs as being directly connected (RT, PP, LH, VM, PF, VAL, IMD, SPA, SPFp, VPMpc, and POL, see the caption of *Figure 1* for definitions of the abbreviations). Third, using the identified thalamic regions as a seed, we found direct projections to other thalamic subregions (MD, CM, AM, VPM, VPL, PO, CL, SPFm, LP, RH, RE, and LD; *Figure 1*). Fourth, using all thalamic ROIs, which were identified in the second and third steps, we identified connected cortical ROIs but only maintained those with significant projections to CP, thereby closing the loop. These cortical regions were ACA, AI, MOp/s, SSp, GU, VISC, PL, RSP, ORB, PERI, VIS, and PTLp (*Figure 1*).

Some of the extracted thalamic and cortical ROIs consisted of only a few voxels and were consequently merged to improve the SNR for the BOLD time-series analysis. The criteria for merging ROIs were that the ROIs (i) were anatomically located next to each other and (ii) had similar anatomical connectivity patterns derived from Allen Brain Atlas. The following groups of thalamic ROIs were merged: (1) RE, LH, and RH; (2) SPF, SPA, and PP; (3) PO and POL; (4) VAL, VPM, and VPMpc; (5) VM and CM; and the following cortical ROIs were merged: (6) VISC and GU and (7) ECT and PERI. This resulted in 14 thalamic (TH) ROIs and 12 cortical (CTX) ROIs.

The mesoscale structural connectome (*Oh et al., 2014*) contains the caudate-putamen (CP) as a single area. Since we neuromodulated the dorsomedial subarea of CP, we further refined the striatal parcellation using more specific anatomical information. Recent research has shown that the CP consists of functionally segregated parts (*Hintiryan et al., 2016*) that can be distinguished along a rostral-caudal gradient, a dorsal-ventral gradient, and a medial-lateral gradient. Using this information from the Mouse Cortico-Striatal Projectome (*Hintiryan et al., 2016*), we further parcellated caudate-putamen into 29 distinct regions.

Finally, we used information from the literature (*Alloway et al., 2017*; *Collins et al., 2018*; *Díaz-Hernández et al., 2018*; *Evangelio et al., 2018*; *Guo et al., 2015*; *Hunnicutt et al., 2016*; *Lee et al.,

*2020*; *Mandelbaum et al., 2019*; *Parker et al., 2016*; *Perry and Mitchell, 2019*; *Wall et al., 2013*) to identify which of the thalamic nuclei from the structural connectome identified above (*Oh et al., 2014*) were specifically anatomically connected to CPdm (our targeted subarea of the caudate putamen). Based on this information, we refined our selection of thalamic ROIs to VM|CM, LD, MD, PF, LP, and CL, which are all reciprocally connected to CPdm, that is, these thalamic nuclei receive projections from CPdm via the GP/SN and project back to the CPdm (*Supplementary file 1*).

In summary, the striato-thalamo-cortical circuit consisted of 14 cortical areas (CTX), 21 thalamic areas (TH), 29 striatal subareas, globus pallidus internal and external (GPi/e), substantia nigra pars reticulata and pars compacta (SNr/c), and subthalamic nucleus (STN).

## Resting-state fMRI data analysis
### Classifying univariate BOLD time series
To understand how cellular-level manipulations shape the BOLD dynamics of structurally well-defined striato-thalamo-cortical circuitry at the macroscale level, time series of each ROI in this circuit were obtained. Each univariate BOLD time series was then represented as a large feature vector, where each feature corresponds to an interpretable summary statistic computed from the time series. Feature extraction was performed using the *hctsa* toolbox, v1.04 (*Fulcher and Jones, 2017*; *Fulcher et al., 2013*), which computes 7702 time-series features per time series. For each ROI, features were computed for each of two time periods (baseline vs. post-clozapine injection) in each of 38 individual subjects (n = 13 D1 excitation, n = 15 D1 inhibition, and n = 10 controls). We filtered out features which were 'poorly behaved,' outputting non-real-values such as NaN, Infinity, or errors. After their removal, regions retained 6588 features out of 7700 that were well-behaved for all regions and belonged to broad categories of distribution, correlation, information theory, model fitting and forecasting, stationarity, nonlinear time-series analysis, Fourier and wavelet transforms, periodicity, statistics from biomedical signal processing, basic statistics, and others.

Feature-based representations of BOLD dynamics in each brain area were used as the basis for classifying different experimental conditions. Because our analysis focused on *changes* of time-series features due to DREADD activation, we subtracted time-series features computed at baseline from those obtained during the post-clozapine period. To classify a given brain area's BOLD dynamics, the subtracted features were normalized using an outlier-robust sigmoidal transformation (*Fulcher et al., 2013*) and passed into a linear support vector machine (SVM). To account for class imbalance ($n_{control}$ = 10; $n_{excitation/inhibition}$ = 13/15), we used inverse probability class reweighting to train the SVM and balanced accuracy (the arithmetic mean of sensitivity and specificity) to evaluate classification performance. Performance was assessed as the mean tenfold stratified cross-validated balanced classification accuracy (BCA), averaged over 50 repetitions (to reduce variance caused by the random partition of data into tenfolds).

In small samples, there is a greater probability that false-positive classification results can be obtained by chance. To account for this effect, the statistical significance of each classification result was estimated using a permutation test. We permuted the assignment of group labels to time series to construct a null distribution of the BCA metric for each region (mean across 50 repeats of tenfold mean cross-validated BCA) for 5000 random group-label assignments to the data. This allowed us to estimate a p-value for each region, which was corrected across all regions by controlling the FDR using the method of Benjamini and Hochberg (*Benjamini and Hochberg, 1995*).

For individual regions with significant classification performance, we also aimed to understand which, if any, features were *individually* discriminative of D1 excitation versus controls. To this end, we scored the discriminability of each feature using a Mann–Whitney rank-sum test and computed corrected p-values across all the 6588 features using FDR correction for each ROI separately.

To account for the fact that our analysis has a much lower number of observations (*n*) than the number of features (*p*), we investigated the potential for overfitting (which would degrade our classification performance) by fitting the same linear regularized SVM model in a lower-dimensional transformation of the original (~7000-dimensional) feature space. We used a simple heuristic of retaining as many principal components of the time series × feature matrix as required to obtain a 100% in-sample accuracy (across training folds of the tenfold cross-validation). This yielded feature spaces of substantially reduced dimensionality, between 4 and 10 (across brain regions). We found no statistical differences between classification performance in the low-dimensional feature space relative to the original,

high-dimensional feature space, consistent with our regularized SVM being minimally affected by overfitting (*Figure 6—figure supplement 1*). We thus present results using the regularized SVM in the full feature space in this work.

## Resting-state fMRI functional connectivity analysis

FC between CPdm and all other ROIs was measured using regularized Pearson's correlation coefficients implemented in FSLnets. A one-sample Wilcoxon test was performed to test whether baseline FC differs from 0 (*Figure 7A*). For each ROI, normalized (post-clozapine minus baseline) connectivity values were entered into a general linear model (GLM) implemented in FSL, and comparisons across the three groups (D1 excitation, D1 inhibition, and D1 control) were performed. Age and weight were used as covariates. Permutation testing with 5000 permutations was performed to estimate whether DREADD activation caused FC changes between groups, followed by the correction for multiple comparisons (for multiple regions and multiple groups) using FDR with a significance threshold, $p < 0.05$. To visualize DREADD-induced changes in FC, baseline connectivity was subtracted from post-clozapine connectivity values such that $\Delta FC > 0$ indicates a DREADD-induced increase in FC, and $\Delta FC < 0$ indicates a DREADD-induced decrease.

### Histological evaluation of transfection

DREADDs viral expression (for DIO-hM3Dq-mCherry, DIO-hM4Di-mCherry, and DIO-mCherry) was confirmed by mCherry staining using standard immunohistochemistry protocols, while qualitative transfection of D1 MSNs was confirmed using antibodies against D1 marker prodynorphin. Briefly, after the last MRI session, mice were deeply anesthetized using a mixture of ketamine (100 mg/kg; Graeub, Switzerland), xylazine (10 mg/kg; Rompun, Bayer), and acepromazine (2 mg/kg; Fatro S.p.A, Italy) and transcardially perfused with 4% paraformaldehyde (PFA, pH = 7.4). The brains were postfixed in 4% PFA for 1.5 hr at 4°C and then placed overnight in 30% sucrose solution. Brains were frozen in a tissue mounting fluid (Tissue-Tek O.C.T Compound, Sakura Finetek Europe B.V., Netherlands) and sectioned coronally in 40-μm-thick slices using a cryostat (MICROM HM 560, histocom AG-Switzerland). Free-floating slices were first permeabilized in 0.2% Triton X-100 for 30 min and then incubated overnight in 0.2% Triton X-100, 2% normal goat serum, guinea pig anti-prodynorphin (1:500, Ab10280, Abcam) and rabbit anti-mCherry (1:1000, Ab167453, Abcam) or rabbit anti-cfos (1:5000, AB2231974, Synaptic Systems) at 4°C under continuous agitation (100 rpm). The next day, sections were incubated for 1 hr in 0.2% Triton X-100, 2% normal goat serum, goat anti-rabbit Alexa Fluor 546 (1:300, A11035, Life Technologies), goat anti-guinea pig Alexa Fluor 647 (1:200, Cat#A-21450, Thermo Fisher Scientific), and DAPI (1:300, Sigma-Aldrich) at room temperature under continuous agitation. Afterward, slices were mounted on the superfrost slides where they were left to air-dry and later coverslipped with Dako Fluorescence mounting medium (Agilent Technologies). A confocal laser-scanning microscope (CLSM 880, Carl Zeiss AG, Germany) and Zeiss Brightfield microscope (Carl Zeiss AG) were used to detect the viral expression. The microscopy protocol included a tile scan with a 10× or 20× objective, pixel size of 1.2 μm and image size of 1024 × 1024 pixels. Images were preprocessed and analyzed using ImageJ-Fiji.

Having obtained confocal images from mice that underwent D1 excitation and D1 inhibition, we further examined whether there were any differences between the number of inhibitory DREADD transfections versus excitatory DREADD transfections of D1 neurons. Five mice were randomly chosen per group. Then, a section covering roughly 200 μm by 800 μm (window based on image size) was drawn over 10× confocal images. We believe our random selection of a 200 μm by 800 μm window for calculating transfected cells can be used as a general reference because it has been shown that D1 neurons are homogeneously spread around dorsal striatum (*Gagnon et al., 2017*). The number of cells (stained using DAPI and mCherry) was manually counted. The percentage of total DAPI cells per region was measured and the results across groups were compared. We found no differences in the number of transfected neurons per chosen ROI across excitatory and inhibitory groups (*Figure 2—figure supplement 1B*). The number of DREADD transfected neurons did not differ significantly across groups.

## Acknowledgements

We thank the team of the EPIC animal facility for providing animal care. We thank Jean-Charles Paterna from the Viral Vector Facility (VVF) of the Neuroscience Center Zurich, a joint competence center of ETH Zurich and University of Zurich for producing viral vectors and viral vector plasmids. This work was supported by ETH Research (grant ETH-38 16-2 to NW), SNSF AMBIZIONE (PZ00P3_173984/1 to VZ), SNSF ECCELLENZA (PCEFP3_203005 to VZ), and SNSF Postdoc.Mobility (214392 to MM). NW is additionally supported by the National Research Foundation, Prime Minister's Office, Singapore, under its Campus for Research Excellence and Technological Enterprise (CREATE) program.

## Additional information

### Funding

| Funder | Grant reference number | Author |
|---|---|---|
| Swiss National Science Foundation | Postdoc.Mobility (214392) | Marija Markicevic |
| Swiss National Science Foundation | Ambizione (PZ00P3_173984/1) | Valerio Zerbi |
| ETH Zürich | ETH-38 16-2 | Nicole Wenderoth |
| Swiss National Science Foundation | Eccellenza (PCEFP3_203005) | Valerio Zerbi |

The funders had no role in study design, data collection and interpretation, or the decision to submit the work for publication.

### Author contributions

Marija Markicevic, Conceptualization, Data curation, Formal analysis, Investigation, Visualization, Methodology, Writing – original draft, Writing – review and editing; Oliver Sturman, Formal analysis; Johannes Bohacek, Resources, Supervision, Writing – review and editing; Markus Rudin, Resources, Writing – review and editing; Valerio Zerbi, Conceptualization, Supervision, Funding acquisition, Writing – review and editing; Ben D Fulcher, Conceptualization, Software, Formal analysis, Supervision, Investigation, Methodology, Writing – review and editing; Nicole Wenderoth, Conceptualization, Resources, Formal analysis, Supervision, Funding acquisition, Investigation, Methodology, Writing – review and editing

### Author ORCIDs

Marija Markicevic https://orcid.org/0000-0003-0983-432X
Johannes Bohacek http://orcid.org/0000-0002-8442-653X
Ben D Fulcher https://orcid.org/0000-0002-3003-4055
Nicole Wenderoth https://orcid.org/0000-0002-3246-9386

### Ethics

All experiments and procedures were conducted following the Swiss federal Ordinance for animal experimentation and approved by Zurich cantonal Veterinary Office (ZH238/15 and ZH062/18).

### Decision letter and Author response

Decision letter https://doi.org/10.7554/eLife.78620.sa1
Author response https://doi.org/10.7554/eLife.78620.sa2

## Additional files

### Supplementary files

• MDAR checklist

• Supplementary file 1. List of thalamic regions of interest (ROIs) and their anatomical projections from/to dorsomedial CP with references to the literature where the information was obtained.

Full names of abbreviated ROIs: CPdm, caudate putamen dorsomedial; SNr, substantia nigra reticular part; GPi, globus pallidus internal; VPL, ventral posterolateral nucleus of thalamus; RE|LH|RH, nucleus of reuniens|lateral habenula|rhomboid nucleus; PO|POL, posterior complex and posterior limiting nucleus of thalamus; SPF|SPA|PP, subparafascicular nucleus subparafascicular area with peripeduncular nucleus of thalamus; IMD, intermediodorsal nucleus of thalamus; RT, reticular nucleus of thalamus; AM, anteromedial nucleus; CL, central lateral nucleus of thalamus; PF, parafascicular nucleus; VAL|VPM|VPMpc, ventral anterior-lateral complex of the thalamus with ventral posteromedial nucleus of the thalamus and its parvicellular part; MD, mediodorsal nucleus of thalamus; LD, lateral dorsal nucleus of thalamus; LP, lateral posterior nucleus of thalamus; VM|CM, ventral and central medial nuclei of thalamus.

• Supplementary file 2. List of all significant features from VM|CM region listed by their names and feature scores in order of significance.

### Data availability

The authors confirm that the data supporting the findings of this study are openly available for download via ETH Research Repository DOI: https://doi.org/10.3929/ethz-b-000539052.

The following dataset was generated:

| Author(s) | Year | Dataset title | Dataset URL | Database and Identifier |
|---|---|---|---|---|
| Markicevic M, Zerbi V, Wenderoth N | 2022 | Neuromodulation of striatal D1 medium spiny neurones measured with resting-state functional MRI - preprocessed dataset | https://doi.org/10.3929/ethz-b-000539052 | ETH Zurich Research Collection, 10.3929/ethz-b-000539052 |

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
