## [Editor Report]

This manuscript provides a valuable set of findings that provide clarity on how striatal direct pathways regulate macroscopic information flow in the brain via the thalamus. The manuscript represents a powerful piece of evidence that will be relevant to researchers across many fields in neuroscience.

---

## [Decision Letter]

**Decision letter after peer review:**

Thank you for submitting your article "Neuromodulation of striatal D1 cells shapes BOLD fluctuations in anatomically connected thalamic and cortical regions" for consideration by *eLife*. Your article has been reviewed by 3 peer reviewers, including Timothy Verstynen as the Reviewing Editor and Reviewer #1, and the evaluation has been overseen by Michael Frank as the Senior Editor.

1. Classifier as an evaluation method.

Reviewer 1 points out that the authors largely rely on a support vector machine (SVM) classifier to predict whether BOLD dynamics within atlas-defined regions reflect stimulation-on or stimulation-off windows. While in one way this is a conservative method for evaluating stimulation effects in the resting BOLD fluctuations, the authors largely report their findings as accuracies of the classifier. Figures 3-5 largely only report model accuracy effects, but we get no sense as to what exactly is happening to the BOLD dynamics in each region. The autocorrelation analysis (Figure 6) somewhat tries to get at this, but only for a subset of regions and the results are largely unclear (see comment below). As a result, a key goal of the study is left largely unaddressed for the reader: i.e. how do intra-region BOLD dynamics change with direct pathway stimulation? The study needs more effort put into this descriptive level of analysis to complement the rigorous classifier analysis.

This reviewer also suggests that the classifier method itself seems highly parameterized. The hctsa method returns 7702 features for each time series. It is unclear exactly how many were in the final set used to run the classification, but even if half of the features were removed, it would still make the classification problem highly overparameterized (e.g., 23 and 25 observations against thousands of features for the excitation and inhibition classifiers respectively). Assuming the authors used cross-validation correctly (which we need more information to support), the risk of inflated classification performance is mitigated. However, we need the details to be able to vet that the bias-variance tradeoff was resolved effectively in this model. In addition, it would be nice to know the features that loaded highly on the final model to resolve the questions about what changes in the local BOLD dynamics from excitation and inhibition of the direct pathway.

Along these same lines, Reviewer 3 points out that, in the first finding the authors show that D1 activation/inactivation produces reliable changes in the infected region (DMS), but most importantly, also produced changes in adjacent areas, suggesting intra-striatal communication. The way the data is presented and discussed appears to be confirmatory of what has been previously described with electrophysiological recordings. In their opinion, the most important part of this section would be to fully describe the differences between activation and inactivation groups. Is interesting that opposite manipulations of D1 receptors produced very similar maps of discrimination (Figure 3). Therefore, it would be necessary to discuss the meaning of obtaining similar classification accuracy indices with opposite manipulations. Perhaps, the use of SVM classifiers can be complemented with other analytical techniques to further disentangle the consequences of manipulating intrastriatal D1 receptors.

2. Variation across individuals

Reviewer 2 points out that while the methods adopted by the authors to acquire the data and evaluate the experimental manipulations are robust and the obtained results are compelling, the current analysis comes short of relating whether variation that can be estimated across the animals has an impact on these results. Specifically, the authors do not leverage the individual animal viral expression or impact on behavior to constrain and estimate the observed responses reported subsequently. Several reports in humans have used individual variability to estimate the relation between behavior and changes in the BOLD fMRI responses at rest, and a basic demonstration of this type of result has been achieved in mice. Applying a similar approach here would further strengthen the result reported here by identifying which regions are linked to the behavioral deficit (e.g., whether the primary motor cortex is linked to contraversive/ipsiversive rotations at the individual level).

This reviewer also highlights that it would be ideal to link the behavior of individual animals to changes in the fMRI signal. This requires an estimation of structure-function that is driven by each individual animal's expression map may enhance the current analysis approach by leveraging potential subtle expression variations to reveal whether the observed changes can be explained by the extent to which expression is different across animals. In addition, a quantification of the difference between the excitatory and inhibitory cohorts will rule out that differences in the impact on the fMRI signal were a result of unintentional group differences in expression extent. The authors should consider adding analyses that relate the individual animal's behavior to the results observed. Namely, whether the magnitude of change in contraversive/ipsiversive rotations is linked to classification-based or pairwise observed results. The authors should consider replicating the analyses conducted in humans to leverage individual variability. This will also strengthen the report by linking its result to human work.

3. Local stimulation effects in the striatum

Reviewer 1 thought that Figure 3 is quite confusing. The classifier is supposed to predict stimulation (excitation or inhibition) on vs. stimulation off (control) periods. This would predict a single number (balanced prediction accuracy) per striatal nuclei. Yet the heat maps shown in this figure show classification accuracies for both stimulation and control conditions. Where do the two numbers come from? Also, given the extremely limited short-range lateral connectivity in the striatum, why are the only stimulation effects observed not in the subnucleus being stimulated (Cpl,dm,cd), but for adjacent subnuclei (CPre, CPivmv) and *only* for excitation conditions? This lack of direct change in BOLD dynamics at the stimulated site seems important and largely ignored.

4. Quantifying viral expression

Reviewer 2 points out that a significant weakness in the current version of the manuscript is the lack of quantification of the viral expression. Currently, the authors do not provide enough information on the extent of coverage of viral expression on average or at the individual level. In particular, while the authors are careful to use the Allen Mouse Brain Connectivity atlas to constrain the fMRI results, they do not relate the specific expression extent, to clearly communicate to the readers, which regions within the striatum are likely to have better representation given the actual expression levels. A better presentation of the extent of the viral expression maps should be provided. The current exposition in Figure 1 is not sufficient to estimate the extent and consistency of expression across animals in each of the groups.

5. Time series effects

Reviewer 1 points out that the one attempt to characterize what happens in the intra-region BOLD dynamics is the autocorrelation analysis reported on page 11 and in Figure 6. However this analysis (a) only focuses on the thalamic nuclei (not also the cortical and the single striatal site shown to exhibit stimulation effects) and (b) only focuses on a few time series measures. Why this limited focus of both target (thalamic nuclei) and measure (first lag of the autocorrelation)? There are many measures for characterizing the temporal characteristics of autocorrelated series from the hctsa analysis. This selective focus seems both narrow and incomplete.

This concern is echoed in Reviewer 3's concern about the observation that thalamic but not cortical regions presented low-frequency fluctuations. What is the meaning of an increase in slow fluctuations? Why did D1 activation (and not inactivation) induced this effect? Are striatal sub-regions also presenting these slow fluctuations?

6. Connectivity results

Reviewer 1 found the changes in functional connectivity as a result of direct pathway stimulation (excitation and inhibition) are both fascinating and limited. There is a clear excitation/inhibition difference in effects, as shown in Figure 7 B-C. However, Figure 7B suggests something different than the change results shown in Figure 7C. It appears that the application of clozapine increases functional connectivity in the control mice (black line Figure 7B). This effect is exaggerated in the inhibition condition, but (most importantly) direct pathway excitation does not really reveal a significant change in the BOLD connectivity patterns. Now this does not change the authors' overall conclusions (connectivity is suppressed with direct pathway excitation relative to control mice), but the nuance of what is happening in the control mice is important for interpretation purposes: direct pathway excitation does not necessarily decrease functional connectivity but does not express the increase in connectivity observed from the application of clozapine. This needs to be elaborated more.

Along the same lines, Reviewer 1 points out that there is an interesting disconnect between the intra-region results and the inter-region (connectivity) results. It is clear that resting BOLD dynamics in thalamic nuclei that project back to the striatum, as well as more unimodal cortical areas, change from direct pathway stimulation in the dorsal caudate. Yet, only one cortical region (MOp) with significant functional connectivity changes overlaps with the set of nuclei that exhibit intra-region BOLD changes. This suggests that local BOLD dynamics and global connectivity are largely disconnected effects. Yet this seems to be largely ignored in the current work. It would be nice to see more analysis, and discussion, of the intra-region and inter-region stimulation effects.

Reviewer 2 points out that the authors do not use their own nor the Allen Institute data to carry out a formal structure-function analysis (following Stafford et al., 2014 PNAS, for example). This is critical since the authors wish to infer on the impact of their manipulation on both cortical and thalamic regions while the precise region in the striatum that they affect is never quantified. A better presentation of the extent of the viral expression maps should be provided. The current exposition in Figure 1 is not sufficient to estimate the extent and consistency of expression across animals in each of the groups. A formal structure-function analysis will significantly strengthen the current report by linking the expression extent (at the individual or cohort level) to anatomically connected regions.

This concern is echoed in Reviewer 3's comment that the second finding (Figure 4) indicates that thalamic regions forming "closed loops" with the striatum were more affected by chemogenetic manipulations. We knew from anatomical studies that the BG are part of anatomically segregated cortico-BG-thalamic loops. Therefore, it would be expected that these anatomical boundaries would somehow limit functional connectivity maps. Here again, we suggest that the manuscript would be improved with further analysis or discussion. For example, it would be interesting to perform further analysis relating the previous section (local striatal connectivity) with this one. In this section, several thalamic nuclei presented higher levels of classification accuracy, but in the previous section, the authors showed that DMS manipulation also produced the same effects in different intrastriatal regions. Therefore, it is not possible to know if the thalamic effects are related to the manipulation of D1 in the DMS or its adjacent regions.

This reviewer also points out that, in the third finding (Figure 5) the authors show that the most "sensitive" cortical regions to the manipulations were classified as "unimodal". This is an interesting result; however, it would be necessary to at least provide further discussion on its potential meaning. It is important to consider that the cortical regions with significant changes, for example, primary sensorimotor cortices, mainly target the dorsolateral, not the dorsomedial striatum. In this context, would it be possible to establish a new analysis to characterize potential correlations between cortical regions and striatal subregions?

Reviewer 3 identifies concerns about the potential changes in functional connectivity (FC) between the striatum and cortical and subcortical regions. Contrary to the results obtained with the SVM-based analytical tool, FC analysis revealed that D1 activation and inactivation produced opposite results, while D1 activation decreased FC in several cortical and subcortical regions, D1 inactivation increased it. While this set of data is clearly described, the implications of these relationships could be further discussed. For example, how do the authors explain that FC with SSp was not significantly changed with this analytical method, but was one of the most affected regions with the Balanced Classification Accuracy method?

Finally, Reviewer 3 points out that in rodents, the dorsal striatum is anatomically and functionally segregated into dorsomedial and dorsolateral, with the last one more related to sensorimotor functions. Please include in the discussion the potential implications of this segregation in the current set of data. Would the authors expect different results if D1 is manipulated in the DLS?

7. Links to behavior

Reviewer 3 noticed that there is no section in the discussion where the behavioral effects observed in figure 2 are contextualized in the massive set of BOLD results presented in the following sections. Do brain-behavior associations exist in this data set?

[Editors' note: further revisions were suggested prior to acceptance, as described below.]

Thank you for resubmitting your work entitled "Neuromodulation of striatal D1 cells shapes BOLD fluctuations in anatomically connected thalamic and cortical regions" for further consideration by *eLife*. Your revised article has been evaluated by Michael Frank (Senior Editor) and a Reviewing Editor.

The manuscript has been improved but there are some remaining issues that need to be addressed, as outlined below:

*Reviewer #1 (Recommendations for the authors):*

The authors have done an excellent job addressing most, but not all, of my original concerns. I have one lingering major concern and one minor issue with the revised manuscript.

1. My original point regarding the interpretation of the classifier results as an evaluation method of the influence of D1 excitation/inhibition on the BOLD response remains somewhat unaddressed. The authors have done an incredible job providing many more details, particularly with the inclusion of the new Figure 6, which highlights how different timeseries features contribute to the classification accuracy. However, this does not really help to address the original question: what is changing with stimulation? In the abstract, the authors conclude "Our results provide a comprehensive understanding of how targeted cellular-level manipulations affect local BOLD dynamics at the macroscale, contributing to ongoing attempts to understand the influence of structure-function relationships in shaping inter-regional communication at subcortical and cortical levels." A similar sentence is made in the Discussion. But I am not entirely sure that this is correct. We know that there are *some* changes in the timeseries characteristics that allow for a classifier to reliably discern stimulation from control conditions, but the complexity of the feature space makes it nearly impossible to interpret the important question of "how". Exactly are the connectivity patterns changing? Is connectivity increasing or decreasing? Is the nature of the autocorrelative structure in the signal increasing or decreasing? Are the specific features that change uniform across regions or do they vary? This remains unanswered in the current version of the manuscript and leaves the reader scratching their head as to what to make of D1 excitation effects, other than to conclude that "something happens."

---

## [Author Response]

Essential revisions:1. Classifier as an evaluation method.Reviewer 1 points out that the authors largely rely on a support vector machine (SVM) classifier to predict whether BOLD dynamics within atlas-defined regions reflect stimulation-on or stimulation-off windows. While in one way this is a conservative method for evaluating stimulation effects in the resting BOLD fluctuations, the authors largely report their findings as accuracies of the classifier. Figures 3-5 largely only report model accuracy effects, but we get no sense as to what exactly is happening to the BOLD dynamics in each region. The autocorrelation analysis (Figure 6) somewhat tries to get at this, but only for a subset of regions and the results are largely unclear (see comment below). As a result, a key goal of the study is left largely unaddressed for the reader: i.e. how do intra-region BOLD dynamics change with direct pathway stimulation? The study needs more effort put into this descriptive level of analysis to complement the rigorous classifier analysis.

We thank the reviewer for pointing out the importance of clarifying the specific BOLD dynamic changes that occur after D1 excitation and D1 inhibition. We address this point by providing further analyses of time-series properties after D1 excitation vs D1 control and D1 inhibition vs D1 control. The text outlined below is a part of the Results section titled “Common time-series properties drive successful classification among thalamic regions.”

“D1 excitation vs D1 control

We analysed the cortical (SSp, PTLp, MOp and VIS), thalamic (PF, LP, LD, MD and VM|CM), and striatal (sub)regions where the classification accuracy (using the full *hctsa* feature set) was statistically significant after. For each of these regions, we determined which *individual* time-series features differ significantly between the D1 excitation vs D1 control group by calculating the signed Mann–Whitney rank-sum statistic and associated *p*-value (for each individual feature) and correcting for multiple comparisons across all features (n=6588) by controlling the false discovery rate (FDR). Repeating the analysis described above for the significant regions (PTLp and SSp) after D1 inhibition was compared to D1 control did not find any individually significant features. Therefore, we focus on the results after D1 excitation was compared to D1 controls and refer to this Mann–Whitney rank-sum statistic as a ‘feature score’ here.

Discriminative individual features were found in two regions: (i) VM|CM (344 significant features) and (ii) LD (284 significant features). These two regions have the highest balanced classification accuracies using the full *hctsa* feature set (Figure 4 – Supplement Figure 1): (i) 90.4% and (ii) 80.3%. First, we focused on VM|CM to assess which types of BOLD time-series properties features best distinguish D1 excitation from D1 control by manually inspecting the list of 344 significant features (Supplement File 2). Overall, we found that these high-performing features were directly measuring, or sensitive to changes in various types of autocorrelation properties of the BOLD signal. To investigate this observation further, we focused on the 100 most discriminative features, which have *p*_FDR_ < 0.01 (note that the patterns are similar when analyzing all 344 significant features). Figure 6A visualizes each of these 100 time-series features as a leaf in a dendrogram, where features with similar behavior on the dataset (measured as the absolute Spearman correlation, |*ρ*|) are placed close to each other. Selected features are annotated onto the dendrogram using colored circles (for a detailed annotation of all features, see Figure 6 – Supplement Figure 2A). The dendrogram reveals a range of time-series features, derived from different types of methods, that exhibit high overall similarity to each other as measured by the Spearman correlation distance (1–|*ρ*| ≤ 0.43) on this dataset: from low time-lag linear autocorrelation coefficients, correlation timescale estimates, state-space and AR model fits, properties of the Fourier power spectrum, automutual information, and fluctuation-analysis methods. Through different specific algorithmic formulations, these features all capture the increase in self-correlation of the signal—consistent with slower and more autocorrelated fMRI signals—in the D1 stimulation condition relative to control. This is most clearly exhibited by the example of lag-1 linear autocorrelation, AC(1), displayed in the Figure 6B-C. Overall, our analysis demonstrates that many individual methods from across time-series analysis—which are sensitive to differences in autocorrelation structure (including low-lag autocorrelations and various autocorrelation timescales)—are useful for distinguishing D1 stimulation from VM|CM fMRI time series.

Next, we aimed to explore whether the same types of BOLD time-series properties vary in broadly similar ways for the LD (for which individual features differed significantly between the D1 excitation and D1 control group); and the remaining regions that exhibited high classification accuracies using the full *hctsa* feature set. In short, we assessed whether changes in BOLD dynamics due to the different stimulation conditions are similar to changes observed in VM|CM by computing correlations between the feature scores (signed rank-sum statistics) for each pair of regions (Figure 6 – Supplement Figure 2B-C). To assess the significance of computed correlation coefficients, we developed a permutation-based method to compute a *p*-value (using 5000 correlated group-label shuffles; i.e., permuting ‘stimulation-on’ and ‘stimulation-off’ labels, while maintaining consistent labelling across the two regions). That is, for each pair of regions, we estimated a null distribution using 5000 group-label shuffles. We found high (ρ > 0.4) and significant correlations between feature scores between VM|CM and other cortical, striatal and thalamic (sub)regions, indicating that D1 excitation causes broadly similar changes to BOLD time-series properties in those regions.

As above, we computed a Mann-Whitney rank-sum test statistic score of each feature for each region and correlated these feature scores with those obtained for VM|CM. The statistical significance of these correlation coefficients was estimated using the permutation test developed above. These *p*-values (11 for the correlation to each other region) were corrected to control the false discovery rate at 0.05. Results indicate that positive correlation coefficients between VM|CM and other thalamic, cortical and striatal regions, which range from ρ = 0.38 – 0.58 are significant (Figure 6 – Supplement Figure 2C). Overall, these results point towards broadly similar changes in time-series properties after D1 excitation in significant thalamic, cortical, and striatal regions.” Investigation of more subtle region-specific differences requires a larger sample size to obtain sufficient statistical power.

Although no individual time-series features were significant in discriminating D1 inhibition from control nor D1 inhibition from D1 excitation, driven by curiosity, similar analysis as described above was performed. Due to the exploratory nature of this analysis, results that follow are only included as a part of this rebuttal. A brief sentence describing results for D1 inhibition and D1 control analysis is mentioned in the main Results section of the paper as follows: ‘Similar analysis as described above did not find any individually significant features following D1 inhibition.’

D1 inhibition vs D1 control

After D1 inhibition, only two regions could be classified on the basis of their BOLD time-series properties: the cortical regions SSp and PTLp. However, no individual time-series features in SSp or PTLp were significant in discriminating D1 inhibition from control (using a Mann-Whitney rank-sum test and controlling FDR). Although the 5% corrected level of statistical significance for inference of individually significant features was not reached, we performed an exploratory analysis to investigate whether the time-series properties that drive significant changes were similar between PTLp and SSp using the same approach as above. We found a significant correlation, ρ = 0.401 (*p*_perm_ = 0.01), indicating a broad similarity in how BOLD time-series properties change in PTLp and SSp after D1 inhibition.

**Author response image 1. sa2fig1:** Pairwise correlation of feature scores of PTLp and SSp (Pearson’s r = 0.405; p_perm_ = 0.01).

D1 inhibition vs D1 excitation

Finally, we explored whether time-series properties change in similar or different ways between D1 excitation and D1 inhibition. For a given target region, we correlated its feature scores both after excitation and after inhibition. We applied this analysis to both SSp and PTLp: the only two regions that could be significantly classified in both the D1 inhibition and D1 excitation conditions. High and significant correlation coefficients were obtained.

**Author response image 2. sa2fig2:** (A) Correlation between feature scores of PTLp after D1 excitation and inhibition, shows high significant positive correlation (Pearson’s r =0.713; p_perm_ < 0.01); (B) Similar to A but for SSp (Pearson’s r=0.788; p_perm_ < 0.01).

Overall, our analyses indicate that similar types of features drive changes within different cortical and thalamic regions when compared between D1 excitation vs D1 control/D1 inhibition vs D1 control/D1 excitation vs D1 inhibition. Features altered after D1 excitation are mostly sensitive to different types of low-order autocorrelation properties (including correlation timescales). In this study, we were only able to illustrate this signature for D1 excitation (and not D1 inhibition); larger sample sizes collected in future, not within the scope of the current article, may allow us to tease apart specific signatures of time-series responses to stimulation, including differences between regions and between excitation and inhibition.

This reviewer also suggests that the classifier method itself seems highly parameterized. The hctsa method returns 7702 features for each time series. It is unclear exactly how many were in the final set used to run the classification, but even if half of the features were removed, it would still make the classification problem highly overparameterized (e.g., 23 and 25 observations against thousands of features for the excitation and inhibition classifiers respectively). Assuming the authors used cross-validation correctly (which we need more information to support), the risk of inflated classification performance is mitigated. However, we need the details to be able to vet that the bias-variance tradeoff was resolved effectively in this model. In addition, it would be nice to know the features that loaded highly on the final model to resolve the questions about what changes in the local BOLD dynamics from excitation and inhibition of the direct pathway.

The reviewer raises important concerns about overfitting classification models in the *p* >> *n* setting, where the number of observations (*n*) is lower than the number of features (*p*). To address this, we have performed several control analyses to show that our results are obtained through out-of-sample cross-validation. These analyses have been added to the Methods section of the paper (Fitting classification models when *p* >> *n*), with new supporting figures added to the supplementary information. Information about which features were significantly altered after D1 excitation are explained in more detail in the Point 1 above.

The control analyses were chosen such that if models were affected by overfitting, it would lead to pessimistic classification results. First, we tested whether there was ‘bleeding’ from training to test sets by calculating the mean of all the classification accuracy null distributions (generated by shuffling input labels) and found that they all are symmetric and centered at the chance-level balanced classification accuracy of 50% (with overfitting expected to shift the mean null accuracy >50%). Combined with direct verification of our code, this test demonstrates that the training and test sets have been properly separated in our cross-validation implementation. An example of a null distribution from a single region is displayed in Figure 6 – Supplement Figure 1 B, D.

Having verified a correct implementation of cross-validation, we respond to the concern that overfitting in a high-dimensional feature spaces may negatively affect our classification results. In this case, our cross-validated results should be improved by instead fitting simpler models in lower-dimensional spaces. Due to the *p* >> *n* setting, our current results were obtained from a very simple regularized linear support vector machine (using a box constraint = 1). To investigate whether these results are underestimating cross-validation performance, we fitted the same linear SVM model in a lower-dimensional transformation of the original (~7000-dimensional) feature space. To do so we, we performed a principal components analysis (PCA) on the time series x feature matrix for each brain region. We then iteratively added leading PCs until the mean in-sample accuracy (i.e., accuracy in training folds of the 10-fold cross-validation procedure) reached 100%, a simple heuristic indicator that the feature space is sufficiently comprehensive (or more-than sufficiently comprehensive) to capture the differences between the labeled classes in a *D*-dimensional feature space. Across different brain regions, *D* ranged between 4–10. We then compared the out-of-sample (mean cross-validation test-fold) accuracy in (i) the full feature space (i.e., our original results), and (ii) the low-dimensional space of *D* leading PCs. Cross-validated accuracies are shown in Figure 6 – Supplement Figure 1A,C. We found no statistical differences between classification models fitted in the original (high-dimensional) feature space, and those fitted in the low-dimensional feature space.

These simple tests thus verified a proper implementation of cross validation, and did not reveal any evidence of over-fitting with our original approach of using a regularized linear SVM in the high-dimensional *hctsa* feature space.

A summary of the analysis is described in the Methods section labeled “Classifying univariate BOLD timeseries”.

“To account for the fact that our analysis has a much lower number of observations (*n*) than number of features (*p*), we investigated the potential for overfitting (which would degrade our classification performance) by fitting the same linear regularized SVM model in a lower-dimensional transformation of the original (~7000-dimensional) feature space. We used a simple heuristic of retaining as many principal components of the time series x feature matrix as required to obtain a 100% in-sample accuracy (across training folds of the 10-fold cross-validation). This yielded feature spaces of substantially reduced dimensionality, between 4 and 10 (across brain regions). We found no statistical differences between classification performance in the low-dimensional feature space relative to the original, high-dimensional feature space, consistent with our regularized SVM being minimally affected by over-fitting (Figure 6 – Supplement Figure 1). We thus present results using the regularized SVM in the full feature space in this work.”

Along these same lines, Reviewer 3 points out that, in the first finding the authors show that D1 activation/inactivation produces reliable changes in the infected region (DMS), but most importantly, also produced changes in adjacent areas, suggesting intra-striatal communication. The way the data is presented and discussed appears to be confirmatory of what has been previously described with electrophysiological recordings. In their opinion, the most important part of this section would be to fully describe the differences between activation and inactivation groups. Is interesting that opposite manipulations of D1 receptors produced very similar maps of discrimination (Figure 3). Therefore, it would be necessary to discuss the meaning of obtaining similar classification accuracy indices with opposite manipulations. Perhaps, the use of SVM classifiers can be complemented with other analytical techniques to further disentangle the consequences of manipulating intrastriatal D1 receptors.2. Variation across individualsReviewer 2 points out that while the methods adopted by the authors to acquire the data and evaluate the experimental manipulations are robust and the obtained results are compelling, the current analysis comes short of relating whether variation that can be estimated across the animals has an impact on these results. Specifically, the authors do not leverage the individual animal viral expression or impact on behavior to constrain and estimate the observed responses reported subsequently. Several reports in humans have used individual variability to estimate the relation between behavior and changes in the BOLD fMRI responses at rest, and a basic demonstration of this type of result has been achieved in mice. Applying a similar approach here would further strengthen the result reported here by identifying which regions are linked to the behavioral deficit (e.g., whether the primary motor cortex is linked to contraversive/ipsiversive rotations at the individual level).This reviewer also highlights that it would be ideal to link the behavior of individual animals to changes in the fMRI signal. This requires an estimation of structure-function that is driven by each individual animal's expression map may enhance the current analysis approach by leveraging potential subtle expression variations to reveal whether the observed changes can be explained by the extent to which expression is different across animals. In addition, a quantification of the difference between the excitatory and inhibitory cohorts will rule out that differences in the impact on the fMRI signal were a result of unintentional group differences in expression extent. The authors should consider adding analyses that relate the individual animal's behavior to the results observed. Namely, whether the magnitude of change in contraversive/ipsiversive rotations is linked to classification-based or pairwise observed results. The authors should consider replicating the analyses conducted in humans to leverage individual variability. This will also strengthen the report by linking its result to human work.

We thank the reviewers for these detailed suggestions. Due to technical reasons that we list in detail below (see Point 4), it has not been possible to obtain a full viral expression for each animal. The modified text is part of both Results and Discussion sections, identically named “Multiple cortical regions display changes in functional connectivity after CPdm neuromodulaton”.

“We further investigated whether there is any relationship between individual behavioral result (post clozapine) and pairwise coupling (FC) between CPdm and ACA, MOp, SSp, and PTLp. These regions were chosen based on: (i) the strength of the FC results (Figure 7C); (ii) strong structural connectivity with CPdm as described by Hintiryan et al. (2016); and (iii) high balanced classification accuracy using *hctsa*. First, we subtracted the number of ipsiversive rotations from the number of contraversive rotations for each animal and used this as a metric of the behavioral effect measured 10 min after subcutaneous clozapine injection. The behavioral metric was then correlated with the animal’s individual FC metric. We found that animals which exhibit strong alterations of FC between CPdm and ACA or MOp (Figure 7 – Supplement Figure 2 A, B) also exhibit strong behavioral changes (*r* ≥ 0.681, *p* < 0.0001). This was only true for cortical regions whose FC is strongly altered after D1 neuromodulation but not for regions with limited FC changes, even if they reached high classification accuracy (Figure 7 – Supplement Figure 2 C, D, *r* ≤ 0.054, p ≥ 0.768). This establishes an association between the post clozapine behavior in an individual mouse and changes in FC between CPdm and the primary motor cortex or anterior cingulate area.

This suggests that these striato-cortical connections are associated with executing coordinated movements (Klaus et al., 2017), a finding in line with a number of reports showing that activity of D1 MSNs in dorsal striatum are necessary for successful execution of locomotion (Badreddine et al., 2022; Barbera et al., 2016).”

3. Local stimulation effects in the striatumReviewer 1 thought that Figure 3 is quite confusing. The classifier is supposed to predict stimulation (excitation or inhibition) on vs. stimulation off (control) periods. This would predict a single number (balanced prediction accuracy) per striatal nuclei. Yet the heat maps shown in this figure show classification accuracies for both stimulation and control conditions. Where do the two numbers come from?

Thank you for drawing our attention to this point. We have clarified this in the updated version of the manuscript and in the new figure caption.

New Main Figure 3B displays the results of classification accuracies obtained between D1 excitation and D1 control (first 2 bars). Separate classifiers were used to obtain classification accuracies for D1 inhibition vs D1 control (new Main Figure 3B, second 2 bars). Therefore, there are two numbers per each striatal subnuclei, which represent results of the two neuromodulation conditions, as compared to controls.

Also, given the extremely limited short-range lateral connectivity in the striatum, why are the only stimulation effects observed not in the subnucleus being stimulated (Cpl,dm,cd), but for adjacent subnuclei (CPre, CPivmv) and *only* for excitation conditions? This lack of direct change in BOLD dynamics at the stimulated site seems important and largely ignored.

We thank the reviewers for pointing this out. Our fMRI dataset does not offer sufficient high spatial resolution to match the detailed CP parcellation obtained via tracer injections. Therefore, we have not fully discussed the classification results of the CP parcellation as this was only an exploratory analysis that needs to be interpreted with caution (Hintiryan et al., 2016).

In the new version of the manuscript, we have completely rewritten that section based on the input of the reviewers. The modified blue text is part of the Results section named “Altered dynamics of virally transfected CP region”.

Cpi,dm,cd is a subnucleus at the exact coordinates of our viral injection site. Its balanced classification accuracy is 75.9% for D1 excitation vs D1 controls and 78.4% for D1 inhibition vs D1 controls. These are high balanced accuracies with significant *p* values after permutation testing. Based on these results, we can conclude that there is a difference in the dynamics of the signal between D1 neuromodulated mice and their controls. However, as shown in the new main Figure 3A, the viral expression could be detected in other subnuclei of the striatum and not just the injection site. This is expected because we injected 950 nl of the virus to ensure sufficient transfection within CPi.

“To test this, we parcellated the CP based on the viral expression maps obtained from all animals. Specifically, we used the CP parcellated atlas containing 29 sub-regions from Hintiryan et al. (2016) and overlaid each with viral DREADD expression maps obtained from each animal. We identified which striatal subregions were transfected with the virus using the criterion that viral expression was detected in at least 1/4 of the CP sub-area in at least two animals. This analysis revealed that Cpi,dm,dl; Cpi,dm,dt; Cpi,dm,im; Cpi,dm,cd; Cpi,vm,vm; Cpi,vl,cvl; Cpr,m; CPr,imd; and CPr,imv have been transfected and these subregions were combined into one region of interest (CPvirus). Next, we extracted BOLD time series from CPvirus and classified controls versus either D1 excitation or D1 inhibition. The balanced classification accuracy for D1 control vs D1 excitation was 75.4% with *p*-value of 0.02, while for D1 control vs D1 inhibition was 71.1% with *p*-value of 0.03 (Figure 3A-B). By contrast, when we repeated the same approach for the non-transfected subareas of the CP (CPoth), the changes in classification accuracy were not significant. This finding confirms that the dorsal striatum (our general target area) was transfected by the virus and affected by excitation and inhibition.”

4. Quantifying viral expressionReviewer 2 points out that a significant weakness in the current version of the manuscript is the lack of quantification of the viral expression. Currently, the authors do not provide enough information on the extent of coverage of viral expression on average or at the individual level. In particular, while the authors are careful to use the Allen Mouse Brain Connectivity atlas to constrain the fMRI results, they do not relate the specific expression extent, to clearly communicate to the readers, which regions within the striatum are likely to have better representation given the actual expression levels. A better presentation of the extent of the viral expression maps should be provided. The current exposition in Figure 1 is not sufficient to estimate the extent and consistency of expression across animals in each of the groups.

We appreciate the reviewers' comments on this point and added new figures and additional analyses to the manuscript (new Main Figure 3, new Figure 2 – Supplement Figure 1). The virus was injected at AP +0.5. Brain slices of 40 μm thickness were cut starting from ~AP +1.0 to ~AP -0.9, which resulted in about 40 slices per brain per mouse. Between 8-10 slices per mouse were stained, taking the first and then every 5th. For our initial analysis, every stained slice was inspected by the experimenter. Viral expression was visible across these slices for all the mice, meaning that the virus spread at least 2 mm following the AP axis. The location of the viral expression for all slices was inspected, spanning approximately 2000 μm of striatum. Individual images of each mouse obtained either using widefield (whole brain slices) or confocal (10 x) microscopy. For clarity, typical whole brain images are presented in Figure 2 – Supplement Figure 1A. Please note that even though every slice was visually inspected, not every widefield image allowed large FOV quantification of viral expression (decision at the time based on (Bay Konig et al., 2019)).

In order to address the reviewers’ comments, we have tried to capture the images of the relevant whole brain slices for each individual mouse but discovered that the brain slices had molded because the fridge had malfunctioned during the first pandemic lockdown in spring 2020. We very much regret that it was impossible to re-image and save the slices, but attempted to address the reviewers’ suggestions based on the data that is still available. Bay Konig and colleagues have recently shown that D1 inhibition at the dorsomedial striatum using DREADDs induces rotational behavior, which is directly proportional to the spatial extent of the viral expression (Bay Konig et al., 2019). Based on this, we decided to use the rotational behavior as our testing mechanism for a successful viral expression of CPdm and only qualitatively assess the extent of mCherry expression. Viral expression displayed against parcellated CP subregions obtained from Hintiryan and colleagues (Hintiryan et al., 2016) is now shown in a newly added Figure 3 displayed in Section 3 above, while analysis on quantifying the viral expression can be found in a Figure 2 – Supplement Figure 1, with the text in the methods section under ‘Histological evaluation of transfection’.

“Having obtained confocal images from mice that underwent D1 excitation and D1 inhibition, we further examined whether there were any differences between the number of inhibitory DREADD transfections vs. excitatory DREADD transfections of D1 neurons. Five mice were randomly chosen per group. Then, a section covering roughly 200 μm by 800 μm (window based on image size) was drawn over 10 x confocal images. We believe our random selection of a 200 μm by 800 μm window for calculating transfected cells can be used as a general reference because it has been shown that D1 neurons are homogeneously spread around dorsal striatum (Gagnon et al., 2017). The number of cells (stained using DAPI and mCherry) was manually counted. The percentage of total DAPI cells per region was measured and the results across groups were compared. We found no differences in the number of transfected neurons per chosen ROI across excitatory and inhibitory groups (Figure 2 – Supplement Figure 1B). The number of DREADD transfected neurons did not differ significantly across groups.”

5. Time series effectsReviewer 1 points out that the one attempt to characterize what happens in the intra-region BOLD dynamics is the autocorrelation analysis reported on page 11 and in Figure 6. However this analysis (a) only focuses on the thalamic nuclei (not also the cortical and the single striatal site shown to exhibit stimulation effects) and (b) only focuses on a few time series measures. Why this limited focus of both target (thalamic nuclei) and measure (first lag of the autocorrelation)? There are many measures for characterizing the temporal characteristics of autocorrelated series from the hctsa analysis. This selective focus seems both narrow and incomplete.This concern is echoed in Reviewer 3's concern about the observation that thalamic but not cortical regions presented low-frequency fluctuations. What is the meaning of an increase in slow fluctuations? Why did D1 activation (and not inactivation) induced this effect? Are striatal sub-regions also presenting these slow fluctuations?

Refer to answers in Point 1, Results section titled “Common time-series properties drive successful classification among thalamic regions” and Discussion section “D1 MSN excitation leads to slower and more autocorrelated fluctuations of the BOLD signal” of the paper.

6. Connectivity resultsReviewer 1 found the changes in functional connectivity as a result of direct pathway stimulation (excitation and inhibition) are both fascinating and limited. There is a clear excitation/inhibition difference in effects, as shown in Figure 7 B-C. However, Figure 7B suggests something different than the change results shown in Figure 7C. It appears that the application of clozapine increases functional connectivity in the control mice (black line Figure 7B). This effect is exaggerated in the inhibition condition, but (most importantly) direct pathway excitation does not really reveal a significant change in the BOLD connectivity patterns. Now this does not change the authors' overall conclusions (connectivity is suppressed with direct pathway excitation relative to control mice), but the nuance of what is happening in the control mice is important for interpretation purposes: direct pathway excitation does not necessarily decrease functional connectivity but does not express the increase in connectivity observed from the application of clozapine. This needs to be elaborated more.

Thank you for this comment. We acknowledge the reviewer’s concerns and performed additional analyses to show that FC is stable in the control mice. The analysis is now a part of Results section, more specifically ‘Multiple cortical regions display changes in functional connectivity after CPdm neuromodulation’.

“To ensure that this significant result is not influenced by FC changes in control animals due to clozapine injection, we re-ran the functional connectivity analysis between CPdm and ACA and MOp without normalization. Figure 7 – Supplement Figure 1 shows raw FC values before and after clozapine injections for each animal (color-coded) of the D1 control, D1 excitation and D1 inhibition group. Most of the control animals retain identical FCs from baseline to post clozapine, while there was a clear change in FC for the D1 excitation and D1 inhibition groups. We performed separate Bayesian paired *t*-tests (JASP, the Netherlands) for each of the six panels in Figure 7 – Supplement Figure 1 to capture changes in FC values from pre to post injection of clozapine. The default settings of JASP yielded robust Bayes factors (BF). For the D1 excitation and D1 inhibition groups, Bayesian *t*-tests revealed BF10s between 3 and 200 indicating that there is moderate to strong evidence in favor of the hypothesis (Jeffreys, 1961) that activating the DREADDs with clozapine caused FC to change. For the control group, by contrast, we tested whether the null hypothesis would hold and obtained BF01 = 0.975 for CPdm-ACA; and BF01=1.043 CPdm-MOp; indicating that there is evidence in favor for the hypothesis that FC did not change from baseline to post clozapine.”

Along the same lines, Reviewer 1 points out that there is an interesting disconnect between the intra-region results and the inter-region (connectivity) results. It is clear that resting BOLD dynamics in thalamic nuclei that project back to the striatum, as well as more unimodal cortical areas, change from direct pathway stimulation in the dorsal caudate. Yet, only one cortical region (MOp) with significant functional connectivity changes overlaps with the set of nuclei that exhibit intra-region BOLD changes. This suggests that local BOLD dynamics and global connectivity are largely disconnected effects. Yet this seems to be largely ignored in the current work. It would be nice to see more analysis, and discussion, of the intra-region and inter-region stimulation effects.

Thank you for this observation. We add further discussion points about this disconnect and the text below can be found in the Discussion section titled “Disconnect between balanced classification accuracy and functional connectivity results”.

“Functional Connectivity (FC) captures the correlation of the BOLD time series *between two regions*, while time-series analysis captures systematic changes in how the BOLD signal fluctuates *within a brain region*. Thus, results obtained with the SVM approach versus the functional connectivity analyses allow researchers to study neural dynamics through two different lenses which provide rich and complementary information on neural dynamics. In this study, the regions with the strongest changes in FC to other regions were not the same as the regions whose local dynamics were most affected by stimulation. Our FC analysis, using Pearson’s correlations between pairs of regions, captures how the strength of their contemporaneous linear statistical association changes after D1 neuromodulation. Studies from our prior work have illustrated rather weak connectivity of thalamic regions with cortical and striatal regions obtained from anesthetized mice (Grandjean et al., 2017), even after cortical neuromodulation (Markicevic et al., 2020). Following this, it is unsurprising that this neuromodulation study does not reveal any FC changes between thalamo-cortical or cortico-thalamic areas. This is one of the reasons we were interested in directly quantifying changes to BOLD signal dynamics in individual regions after D1 neuromodulation. One example of the added value of feature-based classification analyses are the results we have obtained for the thalamus using *hctsa* approach. The properties of the local BOLD signal are conceptually distinct from how they couple to other regions (the alignment of their fluctuations through time) [although they are not strictly independent, e.g., see (Afyouni et al., 2019; Cliff et al., 2021) for an account of how the autocorrelation properties of signals affect the resulting null distribution of Pearson correlation coefficients]. This suggests the importance of assessing brain function and dynamics using multiple approaches.”

Reviewer 2 points out that the authors do not use their own nor the Allen Institute data to carry out a formal structure-function analysis (following Stafford et al., 2014 PNAS, for example). This is critical since the authors wish to infer on the impact of their manipulation on both cortical and thalamic regions while the precise region in the striatum that they affect is never quantified. A better presentation of the extent of the viral expression maps should be provided. The current exposition in Figure 1 is not sufficient to estimate the extent and consistency of expression across animals in each of the groups. A formal structure-function analysis will significantly strengthen the current report by linking the expression extent (at the individual or cohort level) to anatomically connected regions.

In the new version of the manuscript, we now include more information on the extent of viral expression across animals and clarify which subregions of the striatum were virally transfected. Please see Point 4 for further details about how we have analyzed the viral expression. In short, due to technical challenges explained in Point 4 above, we could not investigate the structure–function relationship as suggested by the reviewers.

This concern is echoed in Reviewer 3's comment that the second finding (Figure 4) indicates that thalamic regions forming "closed loops" with the striatum were more affected by chemogenetic manipulations. We knew from anatomical studies that the BG are part of anatomically segregated cortico-BG-thalamic loops. Therefore, it would be expected that these anatomical boundaries would somehow limit functional connectivity maps. Here again, we suggest that the manuscript would be improved with further analysis or discussion. For example, it would be interesting to perform further analysis relating the previous section (local striatal connectivity) with this one. In this section, several thalamic nuclei presented higher levels of classification accuracy, but in the previous section, the authors showed that DMS manipulation also produced the same effects in different intrastriatal regions. Therefore, it is not possible to know if the thalamic effects are related to the manipulation of D1 in the DMS or its adjacent regions.

We acknowledge Reviewer’s suggestion. We have now removed the classification analysis in striatal subregions based on Hintiryan’s atlas (2016) from the manuscript to not over-interpret our results. We have also add new classification results which indicate that viral CP parcellation is a better indicator of explaining how thalamic effects are related to the manipulation of D1 in CPdm. Specifically, most of the striatal subregions that have a high classification accuracy are also transfected by the virus and belong to the dorsomedial striatum (main Figure 3), for details see our response to the Point 4 above. This means that collectively stimulating D1 within these subregions that form the CPdm resulted in altered dynamics within the thalamic regions with direct anatomical projections to CPdm. Additionally, we mentioned in the discussion (Limitation section) that our methods do not allow us to investigate thalamo-striatal connectivity at the sub-nucleus level. This is one of the reasons why we refrain from further analyses involving striatal subregions parcellated based on tracer injections (Hintiryan et al., 2016).

This reviewer also points out that, in the third finding (Figure 5) the authors show that the most "sensitive" cortical regions to the manipulations were classified as "unimodal". This is an interesting result; however, it would be necessary to at least provide further discussion on its potential meaning. It is important to consider that the cortical regions with significant changes, for example, primary sensorimotor cortices, mainly target the dorsolateral, not the dorsomedial striatum. In this context, would it be possible to establish a new analysis to characterize potential correlations between cortical regions and striatal subregions?

We thank the reviewers for these suggestions. The reviewer is right that primary motor and somatosensory cortex project mainly to the dorsolateral striatum. However, this segregation does not seem to hold for striato-thalamo-cortical projections as illustrated in new Figure 4 – Supplement Figure 2. More specifically, most of the thalamic nuclei that are reciprocally connected to the dorsomedial striatum have dense projections to primary motor and somatosensory cortex. We amended the discussion accordingly (please see “Cortical regions with altered BOLD dynamics are primarily unimodal” of the discussion), as well as below.

“Diverse measurements of cortical microstructural properties form macroscopic gradients, argued to shape brain dynamics and function (Parkes et al., 2022; Siegle et al., 2021; Wang, 2020, 2022). For example, Burt and colleagues (2018) show that parvalbumin (PV+) neurons are twice as abundant in unimodal areas than association/transmodal areas, while the opposite is true for somatostatin (SST) neurons (Burt et al., 2018; Wang, 2020). Kim and colleagues (2017) have demonstrated that PV/SST cell densities strongly vary across a proposed hierarchy of mouse cortical areas, with sensory areas containing a greater proportion of putatively output-modulating PV neurons and, conversely, transmodal areas containing a greater proportion of putatively input-modulating SST neurons. This is consistent with the hypothesis that unimodal regions are more involved in feed-forward information streams that require more output-modulating SST neurons than input-modulating PV neurons. Some supporting evidence for this hypothesis comes from Fulcher et al. (2019) who found a weak but significant relationship between T1w:T2w (a candidate proxy for hierarchical level [(Burt et al., 2018)]) and the total incoming structural projection strength to a cortical area, with a lower aggregated input weight to unimodal compared to transmodal areas. In this emerging picture of unimodal areas as being involved in fast processing of sensory inputs and receiving relatively fewer inputs than transmodal areas, a constant perturbation would be predicted to have a greater effect on the dynamics of a unimodal area (for which it constitutes a larger perturbation relative to the low baseline level of input) than a transmodal area (for which it constitutes a relatively small perturbation to the large baseline level of input). This general prediction is supported by our results, in particular a very strong correlation with the response to striatal stimulation across the cortical hierarchy (ρ > 0.7). That is, we found that the change in BOLD dynamics is strongest in unimodal areas and becomes progressively more subtle along the hierarchy. However, given the range of characteristics that vary across cortical areas and may play a role in shaping regional dynamics (Shafiei et al., 2023; Shafiei et al., 2020), future investigations will be crucial in uncovering the relationship between diverse properties of cortical structure and their influence in shaping macroscale neural dynamics of individual cortical areas.

The change in time-series properties observed in cortical areas low in the hierarchy (putatively ‘unimodal’) is an intriguing result, particularly because primary motor (MOp) and somatosensory cortex (SSp) have been shown to project to the dorsolateral (CPdl) rather than the dorsomedial striatum (CPdm). We argue that these changes are likely to be driven via a striato-thalamo-cortical route. As discussed above, dorsomedial but not the dorsolateral striatum has dense, reciprocal connections to the identified thalamic nuclei. Importantly, nearly all of these nuclei project to the unimodal regions, including MOp and SSp, as summarized in Figure 4 – Supplement Figure 2. Thus, striato-thalamo-cortical projections do not seem to exhibit the same level of segregation as cortico-striatal projections and we argue that the effect of our experimental manipulation on time-series dynamics has propagated via this pathway. It is therefore likely that intrinsic properties of unimodal cortical regions result in an increased sensitivity of their BOLD signal to these striato-thalamic inputs.”

Reviewer 3 identifies concerns about the potential changes in functional connectivity (FC) between the striatum and cortical and subcortical regions. Contrary to the results obtained with the SVM-based analytical tool, FC analysis revealed that D1 activation and inactivation produced opposite results, while D1 activation decreased FC in several cortical and subcortical regions, D1 inactivation increased it. While this set of data is clearly described, the implications of these relationships could be further discussed. For example, how do the authors explain that FC with SSp was not significantly changed with this analytical method, but was one of the most affected regions with the Balanced Classification Accuracy method?

We thank the reviewer for this comment and outline that FC and time-series analysis are very different approaches. Discussions on this point can be found in “Cortical regions with altered BOLD dynamics are primarily unimodal” (also part of point 6.5), while the text below can be found in the Discussion section titled “Disconnect between balanced classification accuracy and functional connectivity results”.

“Functional Connectivity (FC) captures the correlation of the BOLD time series *between two regions*, while time-series analysis captures systematic changes in how the BOLD signal fluctuates *within a brain region*. Thus, results obtained with the SVM approach versus the functional connectivity analyses allow researchers to study neural dynamics through two different lenses which provide rich and complementary information on neural dynamics. In this study, the regions with the strongest changes in FC to other regions were not the same as the regions whose local dynamics were most affected by stimulation. Our FC analysis, using Pearson’s correlations between pairs of regions, captures how the strength of their contemporaneous linear statistical association changes after D1 neuromodulation. Studies from our prior work have illustrated rather weak connectivity of thalamic regions with cortical and striatal regions obtained from anesthetized mice (Grandjean et al., 2017), even after cortical neuromodulation (Markicevic et al., 2020). Following this, it is unsurprising that this neuromodulation study does not reveal any FC changes between thalamo-cortical or cortico-thalamic areas. This is one of the reasons we were interested in directly quantifying changes to BOLD signal dynamics in individual regions after D1 neuromodulation. One example of the added value of feature-based classification analyses are the results we have obtained for the thalamus. The properties of the local BOLD signal are conceptually distinct from how they couple to other regions (the alignment of their fluctuations through time) [although they are not strictly independent, e.g., see (Afyouni et al., 2019; Cliff et al., 2021) for an account of how the autocorrelation properties of signals affect the resulting null distribution of Pearson correlation coefficients]. This suggests the importance of assessing brain function and dynamics using multiple approaches.

It is currently unclear which biological mechanism(s) might underpin changes in FC versus changes in time-series dynamics as detected by our classification analyses, but one potential speculative explanation is that FC changes align with information transfer from one area to another, while changes in time-series dynamics might reflect how one area influences neural processing of another which might be sensitive to the effect of neuromodulators.”

Finally, Reviewer 3 points out that in rodents, the dorsal striatum is anatomically and functionally segregated into dorsomedial and dorsolateral, with the last one more related to sensorimotor functions. Please include in the discussion the potential implications of this segregation in the current set of data.

We thank the reviewer for this suggestion. We have now modified the limitation section of the discussion to explain this seeming inconsistency in more detail. “Tracer studies indicate anatomically segregated cortico-striatal projections (Foster et al., 2021; Harris et al., 2019). While it is generally true that dorsal striatum is anatomically segregated into a dorsomedial and dorsolateral portion which receive projections from distinct cortical targets, recent studies using optogenetics have shown that this segregation may not be as strict as initially thought when experimentally stimulating dorsomedial versus dorsolateral portions of the striatum (Grimm et al., 2021; Lee et al., 2016). For example, Lee and colleagues (Lee et al., 2016) optogenetically activated D1 MSNs in CPdm (i.e., the same target area as in the present study) but observed strong functional activity in cortical sensorimotor areas. This discrepancy might be explained by striato-thalamo-cortical projection where recent studies have shown that CPdm projects to specific thalamic nuclei which, in turn, project to unimodal cortical areas including primary motor and somatosensory cortex (see Figure 4 – Supplement Figure 2).”

Would the authors expect different results if D1 is manipulated in the DLS?

Even though we can only speculate about these questions, our findings suggest that neuromodulating D1 neurons in the dorsolateral striatum is likely to result in significant changes in time-series properties within thalamic nuclei which directly project to DLS such as VPM/VPL, while the cortical response would possibly be still tied to unimodal regions such as MOp and SSp. Please see our previous answer for a more detailed discussion, which has also been added to the paper.

7. Links to behaviorReviewer 3 noticed that there is no section in the discussion where the behavioral effects observed in figure 2 are contextualized in the massive set of BOLD results presented in the following sections. Do brain-behavior associations exist in this data set?

Thank you for this point. We have adjusted the discussion and added some analysis accordingly. Please refer to Point 2 above for a more detailed answer.

References

Badreddine, N., Zalcman, G., Appaix, F., Becq, G., Tremblay, N., Saudou, F., Achard, S., & Fino, E. (2022). Spatiotemporal reorganization of corticostriatal networks encodes motor skill learning. *Cell Reports*, *39*(1), 110623. https://doi.org/https://doi.org/10.1016/j.celrep.2022.110623

Barbera, G., Liang, B., Zhang, L., Gerfen, Charles R., Culurciello, E., Chen, R., Li, Y., & Lin, D.-T. (2016). Spatially Compact Neural Clusters in the Dorsal Striatum Encode Locomotion Relevant Information. *Neuron*, *92*(1), 202-213. https://doi.org/https://doi.org/10.1016/j.neuron.2016.08.037

Bay Konig, A., Ciriachi, C., Gether, U., & Rickhag, M. (2019). Chemogenetic Targeting of Dorsomedial Direct-pathway Striatal Projection Neurons Selectively Elicits Rotational Behavior in Mice. *Neuroscience*, *401*, 106-116. https://doi.org/10.1016/j.neuroscience.2019.01.013

Burt, J. B., Demirtas, M., Eckner, W. J., Navejar, N. M., Ji, J. L., Martin, W. J., Bernacchia, A., Anticevic, A., & Murray, J. D. (2018). Hierarchy of transcriptomic specialization across human cortex captured by structural neuroimaging topography. *Nat Neurosci*, *21*(9), 1251-1259. https://doi.org/10.1038/s41593-018-0195-0

Foster, N. N., Barry, J., Korobkova, L., Garcia, L., Gao, L., Becerra, M., Sherafat, Y., Peng, B., Li, X., Choi, J. H., Gou, L., Zingg, B., Azam, S., Lo, D., Khanjani, N., Zhang, B., Stanis, J., Bowman, I., Cotter, K.,... Dong, H. W. (2021). The mouse cortico-basal ganglia-thalamic network. *Nature*, *598*(7879), 188-194. https://doi.org/10.1038/s41586-021-03993-3

Gagnon, D., Petryszyn, S., Sanchez, M. G., Bories, C., Beaulieu, J. M., De Koninck, Y., Parent, A., & Parent, M. (2017). Striatal Neurons Expressing D(1) and D(2) Receptors are Morphologically Distinct and Differently Affected by Dopamine Denervation in Mice. *Sci Rep*, *7*, 41432. https://doi.org/10.1038/srep41432

Grandjean, J., Zerbi, V., Balsters, J. H., Wenderoth, N., & Rudin, M. (2017). Structural Basis of Large-Scale Functional Connectivity in the Mouse. *J Neurosci*, *37*(34), 8092-8101. https://doi.org/10.1523/jneurosci.0438-17.2017

Grimm, C., Frässle, S., Steger, C., von Ziegler, L., Sturman, O., Shemesh, N., Peleg-Raibstein, D., Burdakov, D., Bohacek, J., Stephan, K. E., Razansky, D., Wenderoth, N., & Zerbi, V. (2021). Optogenetic activation of striatal D1R and D2R cells differentially engages downstream connected areas beyond the basal ganglia. *Cell Rep*, *37*(13), 110161. https://doi.org/10.1016/j.celrep.2021.110161

Harris, J. A., Mihalas, S., Hirokawa, K. E., Whitesell, J. D., Choi, H., Bernard, A., Bohn, P., Caldejon, S., Casal, L., Cho, A., Feiner, A., Feng, D., Gaudreault, N., Gerfen, C. R., Graddis, N., Groblewski, P. A., Henry, A. M., Ho, A., Howard, R.,... Zeng, H. (2019). Hierarchical organization of cortical and thalamic connectivity. *Nature*, *575*(7781), 195-202. https://doi.org/10.1038/s41586-019-1716-z

Hintiryan, H., Foster, N. N., Bowman, I., Bay, M., Song, M. Y., Gou, L., Yamashita, S., Bienkowski, M. S., Zingg, B., Zhu, M., Yang, X. W., Shih, J. C., Toga, A. W., & Dong, H. W. (2016). The mouse cortico-striatal projectome. *Nat Neurosci*, *19(8)*, 1100-1114. https://doi.org/10.1038/nn.4332

Hunnicutt, B. J., Jongbloets, B. C., Birdsong, W. T., Gertz, K. J., Zhong, H., & Mao, T. (2016). A comprehensive excitatory input map of the striatum reveals novel functional organization. *ELife*, *5*. https://doi.org/10.7554/*eLife*.19103

Hunnicutt, B. J., Long, B. R., Kusefoglu, D., Gertz, K. J., Zhong, H., & Mao, T. (2014). A comprehensive thalamocortical projection map at the mesoscopic level. *Nature Neuroscience*, *17*(9), 1276-1285. https://doi.org/10.1038/nn.3780

Jeffreys, H. (1961). *Theory of Probability* (Vol. 3rd Edition). Oxford.

Kim, Y., Yang, G. R., Pradhan, K., Venkataraju, K. U., Bota, M., Garcia Del Molino, L. C., Fitzgerald, G., Ram, K., He, M., Levine, J. M., Mitra, P., Huang, Z. J., Wang, X. J., & Osten, P. (2017). Brain-wide Maps Reveal Stereotyped Cell-Type-Based Cortical Architecture and Subcortical Sexual Dimorphism. *Cell*, *171*(2), 456-469 e422. https://doi.org/10.1016/j.cell.2017.09.020

Klaus, A., Martins, G. J., Paixao, V. B., Zhou, P., Paninski, L., & Costa, R. M. (2017). The Spatiotemporal Organization of the Striatum Encodes Action Space. *Neuron*, *95*(5), 1171-1180.e1177. https://doi.org/https://doi.org/10.1016/j.neuron.2017.08.015

Lee, H. J., Weitz, A. J., Bernal-Casas, D., Duffy, B. A., Choy, M., Kravitz, A. V., Kreitzer, A. C., & Lee, J. H. (2016). Activation of Direct and Indirect Pathway Medium Spiny Neurons Drives Distinct Brain-wide Responses. *Neuron*, *91*(2), 412-424. https://doi.org/10.1016/j.neuron.2016.06.010

Leow, Y. N., Zhou, B., Sullivan, H. A., Barlowe, A. R., Wickersham, I. R., & Sur, M. (2022). Brain-wide mapping of inputs to the mouse lateral posterior (LP/Pulvinar) thalamus–anterior cingulate cortex network [https://doi.org/10.1002/cne.25317]. *Journal of Comparative Neurology*, *530*(11), 1992-2013. https://doi.org/https://doi.org/10.1002/cne.25317

Lyamzin, D., & Benucci, A. (2019). The mouse posterior parietal cortex: Anatomy and functions. *Neuroscience Research*, *140*, 14-22. https://doi.org/https://doi.org/10.1016/j.neures.2018.10.008

Markicevic, M., Fulcher, B. D., Lewis, C., Helmchen, F., Rudin, M., Zerbi, V., & Wenderoth, N. (2020). Cortical Excitation:Inhibition Imbalance Causes Abnormal Brain Network Dynamics as Observed in Neurodevelopmental Disorders. *Cerebral Cortex*, *30*(9), 4922-4937. https://doi.org/10.1093/cercor/bhaa084

Parker, P. R. L., Lalive, A. L., & Kreitzer, A. C. (2016). Pathway-Specific Remodeling of Thalamostriatal Synapses in Parkinsonian Mice. *Neuron*, *89*(4), 734-740. https://doi.org/https://doi.org/10.1016/j.neuron.2015.12.038

Parkes, L., Kim, Z. J., J., S., M., C. E., Cieslak M., Gur, R. E., Gur, R. C., Moore, M. T., M., O., Roalf, R., D., Shinohara, T. R., Wolf, H. D., T., S. D., & Bassett, D. S. (2022). Asymmetric signaling across the hierarchy of cytoarchitecture within the human connectome. *Science Advances*, *8*.

Perry, B. A. L., & Mitchell, A. S. (2019). Considering the Evidence for Anterior and Laterodorsal Thalamic Nuclei as Higher Order Relays to Cortex [10.3389/fnmol.2019.00167]. *Frontiers in Molecular Neuroscience*, *12*, 167. https://www.frontiersin.org/article/10.3389/fnmol.2019.00167

Shafiei, G., Fulcher, B. D., Voytek, B., Satterthwaite, T. D., Baillet, S., & Misic, B. (2023). Neurophysiological signatures of cortical micro-architecture. *BioRxiv*. https://doi.org/10.1101/2023.01.23.525101

Shafiei, G., Markello, R. D., Vos de Wael, R., Bernhardt, B. C., Fulcher, B. D., & Misic, B. (2020). Topographic gradients of intrinsic dynamics across neocortex. *ELife*, *9*, e62116. https://doi.org/10.7554/*eLife*.62116

Siegle, J. H., Jia, X., Durand, S., Gale, S., Bennett, C., Graddis, N., Heller, G., Ramirez, T. K., Choi, H., Luviano, J. A., Groblewski, P. A., Ahmed, R., Arkhipov, A., Bernard, A., Billeh, Y. N., Brown, D., Buice, M. A., Cain, N., Caldejon, S.,... Koch, C. (2021). Survey of spiking in the mouse visual system reveals functional hierarchy. *Nature*, *592*(7852), 86-92. https://doi.org/10.1038/s41586-020-03171-x

Wang, X.-J. (2020). Macroscopic gradients of synaptic excitation and inhibition in the neocortex. *Nature* https://doi.org/10.1038/s41583-020-0262-x

Wang, X.-J. (2022). Theory of the Multiregional Neocortex: large-scale neural dynamics and distributed cognition. *Annu Rev Neuroscience* https://doi.org/10.1146/annurev-neuro-110920-035434

[Editors' note: further revisions were suggested prior to acceptance, as described below.]

The manuscript has been improved but there are some remaining issues that need to be addressed, as outlined below:Reviewer #1 (Recommendations for the authors):The authors have done an excellent job addressing most, but not all, of my original concerns. I have one lingering major concern and one minor issue with the revised manuscript.1. My original point regarding the interpretation of the classifier results as an evaluation method of the influence of D1 excitation/inhibition on the BOLD response remains somewhat unaddressed. The authors have done an incredible job providing many more details, particularly with the inclusion of the new Figure 6, which highlights how different timeseries features contribute to the classification accuracy. However, this does not really help to address the original question: what is changing with stimulation? In the abstract, the authors conclude "Our results provide a comprehensive understanding of how targeted cellular-level manipulations affect local BOLD dynamics at the macroscale, contributing to ongoing attempts to understand the influence of structure-function relationships in shaping inter-regional communication at subcortical and cortical levels." A similar sentence is made in the Discussion. But I am not entirely sure that this is correct. We know that there are *some* changes in the timeseries characteristics that allow for a classifier to reliably discern stimulation from control conditions, but the complexity of the feature space makes it nearly impossible to interpret the important question of "how". Exactly are the connectivity patterns changing? Is connectivity increasing or decreasing? Is the nature of the autocorrelative structure in the signal increasing or decreasing? Are the specific features that change uniform across regions or do they vary? This remains unanswered in the current version of the manuscript and leaves the reader scratching their head as to what to make of D1 excitation effects, other than to conclude that "something happens."

We thank the reviewer for further suggestions. Below, we list reviewer’s questions, provide answers and clarify how the article has been modified.

what is changing with stimulation?Is the nature of the autocorrelative structure in the signal increasing or decreasing?Are the specific features that change uniform across regions or do they vary?

We have made further revisions to clarify these points.

First, we have edited the Results section to emphasize that exciting striatal D1 MSN causes an *increase* of the level of self-correlation of the BOLD time series, which can be measured via many different types of algorithms, ranging from linear and nonlinear autocorrelation coefficients, automutual information, statistics of the Fourier power spectrum, and others, as shown in Figure 6A. The figure further demonstrates that all of these features behave similarly on the data and can be interpreted together as pointing to a common change in the fMRI signal in the stimulation condition: an increase in its self-correlation structure (or ‘predictability’). This can be seen in the specific example of the lag-1 autocorrelation, shown in Figures 6B,C.

Second, our results indicate a high and significant correlation (using a permutation test that properly accounts for inter-region dependencies) between the feature scores in VM|CM and other thalamic, cortical, and striatal regions (Figure 6 – Supplement Figure 2). This result points to a broadly common time-series signature of how stimulation affects the BOLD dynamics of these regions. To further illustrate this point we added a new figure as Figure 6 – Supplement Figure 2D showing that the autocorrelation at lag 1 increases for various thalamic and cortical regions. Please see the Results sections titled “Common time-series properties drive successful classification among thalamic regions”:

“A specific illustration of this finding is in Figure 6 – Supplement Figure 2D explicitly showing the increase in autocorrelation at lag 1 after stimulation for the other thalamic and cortical regions.”

Third, we provide a simple explanation for the most prominent change in the BOLD time series in the Discussion section named “D1 MSN excitation leads to slower and more autocorrelated BOLD signal fluctuations” as reproduced below:

“The time-series features responsible for successful classification indicate an increase in self-correlation structure in the fMRI signal. Thorough analyses of individual features that distinguish the stimulation condition ranged from linear to nonlinear autocorrelation coefficients, automutual information, statistics of Fourier power spectrum, and others. Our analyses show that these features were correlated to different types of autocorrelation properties, pointing towards a common change in fMRI signal: an increase in its self-correlation structure. These results are broadly consistent across regions constituting striato-thalamo-cortical circuit.”

Finally, the reviewer is correct in pointing out that the feature space is complex and difficult to comprehend. Accordingly, we have toned down our statement cited by the reviewer above and modified the abstract and Discussion sections accordingly:

“Our results show that targeted cellular-level manipulations affect local BOLD dynamics at the macroscale, such as by making BOLD dynamics more predictable over time by increasing its self-correlation structure. This contributes to ongoing attempts to understand the influence of structure–function relationships in shaping inter-regional communication at subcortical and cortical levels."

How exactly are the connectivity patterns changing?Is connectivity increasing or decreasing?

Regarding functional connectivity changes after D1 excitation, we observed a decrease in FC when D1 excitation group was compared to controls, only in cortical regions i.e., anterior cingulate cortex (ACA) and primary motor cortex (MOp) (Figure 7C). Patterns of FC decrease are present in other cortical regions, although none of these changes were statistically significant. Furthermore, no significant changes were observed in striatal nor thalamic regions. Further discussions can be found in Results and Discussion sections titled, “Multiple cortical regions display changes in functional connectivity after CPdm neuromodulation” and “Disconnect between balanced classification accuracy and functional connectivity results”.